# Triathlon: Ergo Nutrition for Training, Competing, and Recovering

**DOI:** 10.3390/nu17111846

**Published:** 2025-05-28

**Authors:** Álvaro Miguel-Ortega, María-Azucena Rodríguez-Rodrigo, Juan Mielgo-Ayuso, Julio Calleja-González

**Affiliations:** 1Faculty of Education, Alfonso X “El Sabio” University (UAX), 28691 Madrid, Spain; 2Regional Ministry of Castilla y León Board of Education, HS Conde Diego Porcelos, 09006 Burgos, Spain; mrodriguezrrod@educa.jcyl.es; 3Faculty of Health Sciences, University of Burgos (UBU), 09001 Burgos, Spain; jfmielgo@ubu.es; 4Physical Education and Sports Department, Faculty of Education and Sport, University of the Basque Country (UPV/EHU), 01007 Vitoria, Spain; julio.calleja.gonzalez@gmail.com; 5Faculty of Kinesiology, University of Zagreb, 10110 Zagreb, Croatia

**Keywords:** triathlon, nutrition, energy demands, recovery, ergogenic aids

## Abstract

Triathlon is a multi-sport event that combines swimming, cycling and running. The distances vary and the physiological demands are high. **Objectives**: This review compiles information on nutritional strategies and ergogenic supplements for triathlon training, competition and recovery. It aims to provide an understanding of the specific challenges and needs of the sport to help triathletes and coaches optimise performance through effective training and nutrition plans. **Methods**: English-language publications were searched using the keywords triathlon, nutrition, recovery and ergogenic aids, alone or in combination, in databases. **Results**: Maintaining good glycogen levels, consuming enough carbohydrates and staying properly hydrated are key to athletic performance, especially for triathletes. Education regarding nutrition, the role of probiotics and supplements, and diet modification for the enhancement of performance and recovery are pivotal considerations. **Conclusions**: Triathletes are at risk of RED-S due to negative energy balance and high fibre/plant protein diets, especially women. Optimising muscle glycogen through tailored diet and training, especially pre- and in-race nutrition, including carbohydrate loading and hydration strategies, is critical. Education is needed to improve post-exercise nutrition, while probiotics and certain supplements may aid performance and recovery. Dietary support is important for resistance training to optimise performance and recovery.

## 1. Introduction

Common event types, ordered by increasing distance, include: Super Sprint (350 m swim, 10 km cycle, 2.5 km run); Sprint (750 m swim, 20 km cycle, 5 km run); Olympic/International (1.5 km swim, 40 km bike, 10 km run); Middle Distance (1.9 km swim, 90 km bike, 21.1 km run); and Ironman (3.86 km swim, 180 km bike, 42.2 km run) [1].

Gaps in the triathlon literature, particularly in the areas of nutrition and supplementation, limit the optimisation of athletes’ performance and health. There is a lack of personalised strategies, limited scientific evidence specific to triathlon, poorly optimised recovery protocols and the use of supplements based on limited evidence or myths. Many athletes struggle with hydration and electrolytes, nutritional planning during competition has been shown to be poor, lack of physiological monitoring limits real-time adjustments, and a general lack of up-to-date nutritional education is perceived.

Triathlon places significant physiological demands [2] on athletes, requiring high aerobic capacity for sustained exercise at moderate to high intensities [2]. Efficient cardiovascular systems and high VO_2max_ [3] are essential for prolonged effort. Resistance and cross-training are necessary for injury prevention, performance enhancement and recovery [4]. Although primarily aerobic, muscular strength is essential for explosive movements, maintaining technique and resisting fatigue, particularly in cycling and running [5].

Triathletes need an efficient anaerobic system [6] to produce short, intense bursts of energy, especially when transitioning or finishing. [7]. Swimming requires stamina and proper technique to minimise water resistance and optimise energy use [8]. Cycling relies on position and pedalling technique to maximise power output and minimise energy expenditure [9]. Running relies on style and biomechanical efficiency to conserve energy [10]. An efficient anaerobic system is required to rapidly deliver energy [11] with different needs during training [12] (optimising performance and adaptation), competition [13] (maximising energy and concentration) and recovery [14] (repairing muscle, replenishing glycogen and reducing inflammation).

Efficient recovery between triathlon sessions is critical to performance [15] due to the physiological demands of the sport. Triathlon places significant energy demands on athletes during both performance and training [16], emphasising the importance of nutrition [17].

Adequate nutrition is essential for maintaining performance [18], health [19] and preventing disease [20], and has a positive impact on physical condition, cognitive performance and mental health [21]. In sport, optimal nutrition improves exercise [22] adaptation and energy expenditure [23]. A high-protein, moderate-carbohydrate diet with nutrient-dense foods may be appropriate during exercise [24]. Individualised nutritional strategies that optimise carbohydrate (CHO) intake based on exercise intensity and duration [25] are essential, as nutritional needs vary throughout the season [26] and require adaptation [27].

Recovery is crucial in triathlon training [28] for injury prevention, adaptation, energy replenishment, hormonal balance and maximising performance. Ergogenic and nutritional supplements [29] are commonly used for recovery, but should be used cautiously under expert [30] supervision as part of a comprehensive nutrition and performance plan. Ergonutrition has been demonstrated to promote health, prevent disease and optimise physical performance; therefore, it is the recommended dietary approach in this case. In the specific context of sports nutrition, the term “ergonutrition” refers to the adaptation of nutritional recommendations to the specific characteristics and demands of the sport. Adequate protein, CHO, fat and micronutrient intake should be achieved prior to supplementation to maximise benefits [31].

Strategic nutritional (including CHO [32], electrolytes [33], amino acids [34], antioxidants [35] and caffeine [36]) and training support [37] improve the health and performance of triathletes by maximising performance and aiding recovery. Sports equipment should be chosen based on individual needs and tested in training prior to competition.

This systematic review of triathlon is essential for analysing the scientific evidence relating to nutrition, recovery and supplementation during training and competition. It will enable us to identify best practices, highlight areas for further research and enhance the performance and well-being of athletes. The review will rigorously compile, evaluate and synthesise all available scientific evidence. This will allow us to identify patterns and gaps in knowledge and make recommendations based on solid evidence. This will make it easier for coaches, athletes and health professionals to make informed decisions. This systematic review′s novelty lies in its integration of the most recent and relevant findings on the specific effects of nutrition, recovery and supplementation on triathletes. Consequently, the review will provide an updated, consolidated overview to guide future research and enhance sports practices in triathlon.

Nutritional and ergonomic support is essential for coaches and trainers to optimise athletic performance [38]. While reviews exist for other sports (basketball [39,40], volleyball [13], cycling [41], swimming [11] or water polo [42]), there is no review that focuses on training, competition and recovery in triathlon. Therefore, the main objectives of this systematic review are: (i) to review the effective nutritional strategies proposed in the literature; (ii) to evaluate the effect of ergonutritional supplements in triathlon based on scientific evidence to improve the recovery of triathletes; and (iii) to determine which of these ergonutritional supplements are effective for the performance and recovery of athletes.

## 2. Materials and Methods

### 2.1. Sources of Information

This article provides a comprehensive and in-depth analysis of the three cornerstones that support optimal performance in triathlon: strategic nutrition, comprehensive recovery, and intelligent supplementation. Beyond a rudimentary description, an exploration will be undertaken of how each of these aspects interacts and complements each other to maximise the potential of the triathlete. The latest research and best practices in nutrition will be analysed, with consideration given to the specific demands of each stage of training and competition. These include pre-training, intra-training, and post-training. The programme will explore a range of recovery strategies, including optimal sleep techniques, stress management, massage therapies and active recovery, which extend beyond the conventional approach of rest. Finally, the controversial realm of supplementation will be examined, with a rigorous examination of the scientific evidence. The objective of this work is to serve as a valuable and practical source of information.

The review was conducted in accordance with the guidelines set out in the PRISMA statement (Preferred Reporting Items for Systematic Reviews and Meta-Analyses) [43].

The PICOS model has been used for the definition of the inclusion criteria above [44] (Figure 1).

This review focused on studies that met the following criteria: (i) participants had to be triathletes, (ii) articles had to examine topics such as nutritional requirements, recovery strategies and ergogenic aids in triathlon, and (iii) study designs had to be non-randomised. Relevant studies were selected for review, excluding the following: (i) studies with participants from other disciplines or with pre-existing conditions, (ii) articles on other sports populations, abstracts, non-peer-reviewed articles and book chapters, and (iii) studies not related to nutritional needs, recovery strategies or ergogenic aids.

Extensive electronic searches were conducted in several scientific literature databases including Web of Science (WOS), SciELO Citation Index, MEDLINE (PubMed), Current Contents Connect, Cochrane Library, KCI-Korean Journal Database, Embase, ICTRP, CT.gov, EBSCO and Scopus, using a series of keywords related to triathlon, nutrition, recovery and ergogenic: ((‘triathlon’ [MeSH terms] OR ‘triathlon’ [All fields]) AND (‘nutrition’ [MeSH terms] OR ‘nutrition’ [All fields]) AND (‘recovery’ [MeSH terms] OR ‘recovery’ [All fields]) OR (‘ergogenic’ [MeSH terms] OR ‘ergogenic’ [All fields])). These keywords were selected based on the opinion of the authors and a review of the literature, and the vocabulary was checked against Medical Subject Headings (MeSH). The search was not limited by publication date but was restricted to studies conducted in humans and written in English. The last search was performed on 7 April 2025. In addition, the references of the articles found were reviewed to analyse, include and discuss the most relevant studies in the field using a snowball sampling strategy [45]. It was registered in the International Prospective Register of Systematic Reviews (PROSPERO) on 13 February 2025 (registration number: CRD420250652029).

### 2.2. Study Selection

The titles and abstracts of the publications were identified using the search strategy described above and collated to identify duplicates (Figure 2). All included studies were retrieved to determine their eligibility and classified as relevant. In addition, the reference section of all relevant articles was examined [46]. Based on the information provided in the full articles, inclusion and exclusion criteria were used to select studies for inclusion in this systematic review. There was no disagreement among the authors about the eligibility of the studies.

### 2.3. Data Extraction

Relevant data from each study were collected, such as source, type of test, population, methods, characteristics of the intervention and significant differences between groups. The selected studies were then grouped into two categories: (a) nutritional requirements and (b) the relationship between these requirements, recovery and performance support. The authors discussed and reached a final agreement to minimise errors in data collection and group formation.

The process of collecting and pooling the data was not straightforward, as some studies had missing information or inconsistencies in their reports. However, a full understanding of the relevant data was achieved through careful review and discussion among the authors. The included studies were selected according to strict criteria, such as the use of randomised controlled trials and the inclusion of a control group, which ensured the quality and reliability of the information. The process of data extraction and grouping played a key role in the analysis, allowing the data to be systematically examined and important patterns and trends in the relationship between energy requirements, quality control, diet and performance to be identified.

### 2.4. Assessing the Quality of Experiments: Risk of Bias

To reach reliable conclusions, the quality of the publications was assessed using the STROBE^®^ (Strengthening the Reporting of Observational Studies in Epidemiology) guidelines, following the guidelines of the Cochrane Collaboration [48]. These STROBE^®^ guidelines for reporting observational studies [49] were also used to carefully assess the potential limitations of the included studies.

The items on the checklist were classified into different areas: random sequence generation (selection bias), allocation concealment (selection bias), blinding of participants and personnel (implementation bias), blinding of outcome assessment (detection bias), incomplete outcome data (attrition bias), selective reporting (reporting bias), and other types of bias. They were classified as ‘low’ if they met the criteria for negligible risk of bias (plausible bias that does not seriously alter the results) or ‘high’ if they met the criteria for high risk of bias (plausible bias that seriously undermines confidence in the results). If the risk of bias was unknown, it was considered ‘uncertain’ (plausible bias that casts some doubt on the results) [48].

The level of evidence of the selected trials was determined using the Oxford Quality Rating System (Table 1), which is widely used worldwide [50,51]. This tool for assessing the quality of clinical trials has become an international standard of reference [52], as it allows the methodological quality of trials to be assessed in an objective and standardised way, which in turn facilitates the interpretation and comparison of the results obtained. This tool is particularly relevant in the context of systematic reviews and meta-analyses, where the assessment of the quality of the included studies is essential for the validity of the conclusions.

## 3. Results

An initial search of the scientific literature identified 1628 articles on triathlon and other keywords terms relevant to this review. Of these, 925 articles that did not address nutrition, recovery or performance enhancement were excluded. A further 590 articles from other areas that did not meet the criteria were also excluded (Figure 2). Ultimately, 35 articles were identified for inclusion in the review (Table 2).

To identify the studies that would be useful for the review, we first established which studies to include based on the inclusion and exclusion criteria. We then conducted an in-depth search of the databases using the appropriate keywords.

Once the relevant studies to be analysed had been obtained, we reviewed aspects such as titles and abstracts to determine which met the established inclusion and exclusion criteria. Studies that did not meet the inclusion criteria were excluded. Those that did, were then reviewed in full to establish whether they addressed all aspects of the criteria. If a study did not meet the criteria upon full review it was discarded. Finally, the studies that met all the criteria were selected and critically analysed at a methodological level to determine their relevance.

Following this thorough review of the existing literature and application of the established criteria, 35 studies that met the established requirements were identified and included in the analysis.

### 3.1. Assessing the Quality of Experiments: Risk of Bias and Levels of Evidence

Two authors independently assessed the methodological quality and risk of bias independently, and disagreements were resolved by a third reviewer according to the guidelines of the Cochrane Collaboration guidelines [48]. The quality of the studies was classified using the following scale: (1) good quality (>14 points, low risk of major bias); (2) acceptable quality (7–4 points, moderate risk of major bias); and (3) poor quality (<7 points, high risk of major bias) [49].

Many of the analysed trials did not adequately explain how they dealt with other types of error, either due to a lack of information or the exclusion of important data. To determine the quality and usefulness of the nutritional and ergonomic advice, we employed a tool designed to evaluate nutrition and performance trials [87] (Figure 3 and Figure 4).

### 3.2. Nutritional Strategies

Since the last century, nutritional strategies have shown that increasing the nutrient intake of triathletes improves their endurance performance, reduces their fatigue levels by lowering their lactic acid levels, and optimises their use of carbohydrates and fats as energy sources [53].

Kimber et al. (2002) [54] reported that 94% of total energy came from CHO. Hourly CHO intake was similar between the sexes and sufficient for maximum muscle utilisation. CHO consumption was significantly higher during cycling than during running, in both absolute terms and in relation to body weight and time. A significant positive correlation was observed between total CHO intake and final time in triathletes. Cox et al. (2004) [61] found that, although both sexes met the guidelines of 2.1–3.0 g/kg, female triathletes consumed more CHO in the morning before the race than male triathletes. CHO intake during the race was lower than recommended.

A previous review [55] showed that adequate carbohydrate (CHO) loading improves endurance and delays fatigue. This suggests that 60–70 g of CHO per hour during competition is required to maintain energy levels. This emphasises the importance of consuming an adequate amount of CHO before and after the event to ensure good performance. A subsequent study confirmed that CHO are the main source of energy during running [63]. This study also found that solid CHO consumption was higher than liquid CHO consumption. Sareban et al. (2016) [66] found that 77.8% of participants experienced gastrointestinal (GI) symptoms when consuming CHO in gel form, compared to none when consuming them in liquid form.

Although the optimal nutritional strategy for triathlons is well-understood, a significant proportion of triathletes, particularly those in the masters category, are unaware of the recommended post-exercise amounts of carbohydrates and proteins [83]. This highlights the need for educational programmes, such as that of Tan et al. (2022) [77], which demonstrated significant improvements in hydration and nutrition. Women were found to be more knowledgeable than men in these areas. Following the programme, participants increased their consumption of fruit and milk. Current recommendations for CHO intake (6 g/kg/day) and energy (40 kcal/kg/day) are not being met [78], with intakes below recommended levels. Many athletes do not adapt their diet according to the training cycle, and only a minority consume CHO prior to competition [85]. While most consume CHO during competition, very few reach the recommended intake of 60 g/h.

As shown by Witkoś et al. (2023) [81], it is clear that nutrition is essential for health and performance in triathlons. In addition to good nutrition, energy availability and balance are also crucial [88]. Therefore, the results of Witkoś et al. (2023) [81] study on low energy availability (LEA) are relevant for athletes and coaches. Mendes et al. (2025) [85] found that, prior to competition, most participants gradually reduced their energy intake and took CHO supplements during the race. However, they did not implement supercompensation strategies or adhere to the recommended CHO supplements amounts per hour. A recent study [16] showed that the average daily macronutrient intake was 53% carbohydrate, 18% protein and 29% fat.

### 3.3. Recovery

According to Frentsos & Baer (1997) [53] and Jeukendrup et al. (2005) [55], nutritional strategies for recovery, such as CHO and protein intake, promote faster recovery. They recommend consuming 1.0–1.2 g/kg of carbohydrate per hour post-exercise. Bentley et al. (2008) [58] add to this by emphasising the importance of protein ingestion and adequate rest for muscle recovery and adaptation, as well as cooling techniques. Various post-exercise recovery strategies [69] include compression garments (effective in reducing muscle damage, but inconsistent on performance), cold water immersion (effective in reducing delayed onset muscle soreness (DOMS) and inflammation but potentially affecting glycogen recovery), adequate sleep (critical for muscle recovery and injury prevention), protein and BCAA supplementation (effective in promoting muscle repair and glycogen resynthesis) and active regeneration (e.g., foam rolling and light pedalling, for which there is limited evidence). Storsve et al. (2020) [72] showed that krill oil supplementation helps maintain higher levels of choline and dimethylglycine, particularly following endurance events, during which serum choline concentrations decrease significantly. A study using seawater suggests that fibroblast growth factor 21 (FGF-21) and brain-derived neurotrophic factor (BDNF), hormones that regulate lipid and carbohydrate metabolism, may influence recovery [76]. FGF-21 stimulates glucose uptake, while BDNF regulates appetite and body weight. Combining CIT and BR may also speed up muscle recovery, though it does not prevent exercise-induced muscle damage (EIMD) [73].

Additionally, adequate hydration with sodium-containing drinks (30–50 mmol/L) [55] are crucial for water absorption and the prevention of hyponatraemia. Both dehydration and overhydration can be dangerous, so sweating and electrolyte loss should be monitored on an individual basis [58]. Aragón-Vela et al. (2024) [84] found that hydration with mineral-rich deep seawater (DSW) significantly preserved isometric muscle strength during exercise.

### 3.4. Ergogenic Supplementation

Ergogenic aids are used to maximise athletic performance in events such as triathlons. One study investigated the effects of consuming organic grape juice (OGJ) [62] over a period of 20 days. It was found that this increased plasma concentrations of polyphenols, which are relevant due to their low bioavailability. Decreases in fasting plasma glucose and increases in serum insulin were also observed. These results suggest that organic red grape juice may improve antioxidant capacity, glucose homeostasis and microcirculation in endurance athletes, thereby reducing the risk of cardiovascular disease. Another study [67] found that taurine did not improve aerobic performance in triathletes, but reduced oxidative stress. However, it has been shown that caffeine, sodium and organic nitrates are supplements that improve triathlon performance [68].

Some of the reviewed studies show that probiotic supplements, which contain microorganisms that maintain or improve the microbiota, could influence disease incidence, health and metabolism. One study [70] investigated the effect of Lactobacillus plantarum PS128 in triathletes and suggested that it helps maintain performance during exercise by modulating inflammation, oxidation and metabolism. However, seawater does not immediately improve performance or reduce perceived exertion when used as an ergogenic supplement [76]. However, it helps to maintain plasma volume, prevents sudden increases in haematocrit and modulates the post-exercise inflammatory and metabolic responses. Burgos et al. (2022) [73] studied the long-term effects of co-supplementing with citrulline and beetroot extract in triathletes. The results showed significant improvements in muscle mass, mesomorphy, lower body strength and performance in the Cooper test. A complementary study by the same authors showed that this combination prevented an increase in cortisol and a decrease in the T/C ratio, while also promoting greater distance covered in the Cooper test after 9 weeks. Recent studies [35] suggest that vitamin C and antioxidant supplementation improve metabolic function, skeletal oxygenation and cardiac function. In contrast, another study by Durkalec et al. (2023) [78], showed that participants did not achieve recommended levels of micronutrients such as vitamin E, folic acid, vitamin C and calcium.

Regarding the use of dietary supplements, Oliveira et al. (2024) [83] found that 90.2% of the Brazilian triathletes surveyed used them. The most used supplements were amino acids (97.2%) and CHO (83.9%). Most athletes take several supplements, with 75% receiving advice from nutritionists. Jiménez-Alfageme et al. (2023) [80] found that 92.2% of Spanish triathletes use sports supplements, primarily in the form of bars, isotonic drinks, gels and caffeine (AIS group A). Total consumption does not differ significantly by gender or level, although national and international triathletes consume more ergogenic aids from group A, with women consuming more iron and men consuming more caffeine.

## 4. Discussion

The main objectives of this systematic review were: (i) to review the effective nutritional strategies proposed in the literature; (ii) to evaluate the effect of ergonutritional supplements on triathlon performance and recovery, based on scientific evidence to improve recovery in triathletes; and (iii) to determine the effectiveness of these supplements for improving performance and recovery in athletes.

The results of the study are compared with those of previous research in order to contextualise the findings within the fields of science and sport. This is particularly relevant in relation to ergogenic recovery in triathlon and highlights the potential benefits of supplements such as OGJ and BCAAs on performance and recovery.

### 4.1. Nutritional Strategies

Studies have shown that increasing nutrient intake significantly improves endurance performance in triathletes [89]. Glucose derived from CHO stores in the muscles (glycogen) and liver is the main source of energy during these activities, with fat acting as a secondary source [90]. The relative use of these substrates varies according to the intensity and duration of exercise [91]. Other important benefits include reduced fatigue due to decreased concentration and improved utilisation of CHOs and fats for energy production [92]. This is supported by the work of Kimber et al. (2002) [54] who found that on average 94.0% of the total energy expended during an Ironman test was derived from this substrate. In events lasting more than two hours, fatigue may be related to a decrease in muscle and liver glycogen stores [93]. Therefore, an effective strategy should focus on maintaining adequate glycogen levels through appropriate CHO intake [26].

If you are looking for a pre-competition strategy, remember that glycogen loading is essential for maximising liver and muscle reserves prior to the event [94]. Studies show that consuming 7–12 g of CHO per kilogram of body weight in the 1–3 days before the competition can significantly improve performance in prolonged tests [95]. However, it is important to accompany this strategy with a gradual reduction in training to avoid additional fatigue.

Eating a carbohydrate-rich meal 3–4 h before the test will help maintain stable blood glucose levels [96]. To avoid GI discomfort, it is advisable to avoid foods high in fat or fibre [86].

During competition, it is recommended that athletes consume between 30 and 60 g of CHO per hour in order to maintain glycogen levels and delay central fatigue [94]. Sources can include sports drinks, gels, bars, and easily digestible fruit.

This finding is consistent with a previous review by Jeukendrup et al. (2005) [55], which indicated that adequate carbohydrate loading improves endurance and delays fatigue, but that adjustments are needed to avoid digestive discomfort. This study recommended a higher dose of carbohydrates (60–70 g/h). However, beyond specific amounts, the results of both studies reinforce the idea that adequate carbohydrate consumption before and after the event is necessary for good performance during the competition [75], albeit in smaller amounts to avoid gastrointestinal discomfort. To this end, it is crucial to experiment with different types and amounts of carbohydrate during long training sessions in order to determine what works best for each individual, taking GI tolerance and preferences into account [97]. In this regard, Sareban et al. (2016) [66] found that 77.8% of participants reported GI problems with CHO gels (GEL), while none reported problems with liquid CHO (LIQ). In terms of consumption, the study by Barrero et al. (2015) [63] found that there was a tendency to consume more solid CHO than liquid CHO.

To accelerate muscle glycogen recovery, it is recommended that CHO are consumed within the first 30 min after exercise [98]. Consuming a combination of CHO and protein (in a ratio of approximately 3:1) also promotes muscle repair [99] (Figure 5).

In addition to the aforementioned macronutrients, antioxidants found in fruits and vegetables have been shown to reduce the oxidative stress caused by intense exercise [100]. In this regard, Lee et al. (2023) [35] indicated that antioxidants are beneficial for metabolic, skeletal, and cardiac function. Conversely, Neubauer et al. (2010) [59] suggested that antioxidants may prevent oxidative damage to DNA. Furthermore, supplements such as omega-3 fatty acids may have beneficial anti-inflammatory effects. Hotfiel et al. (2019) [69] indicated that they promote muscle repair and glycogen resynthesis.

In terms of dietary strategies, it would be useful to consider the following specific factors:Individualisation: Each athlete has different metabolic needs, GI tolerances and cultural or personal preferences [101].Previous practice: All strategies should be tested during long training sessions to avoid surprises in competition.Environmental factors: Altitude, temperature and humidity affect water and energy requirements. For example, hot environments increase water loss.

### 4.2. Recovery

Optimising athletic performance and promoting muscle adaptation requires careful consideration of post-exercise nutritional recovery [102]. Research by Frentsos and Baer (1997) [53] demonstrated the synergistic effect of consuming carbohydrates and protein together in accelerating recovery. This finding highlights the importance of adopting a combined nutritional strategy rather than focusing solely on a single macronutrient.

Building on the post-competition phase introduced in the previous section, the recovery process in triathlon involves complex mechanisms that restore physiological functions altered by training or competition [15]. These processes include the following:-Muscle repair: Intense activity can cause microlesions in muscle fibres, particularly following high-intensity or high-volume sessions [103]. The repair process requires protein synthesis, controlled inflammation, and the removal of residual metabolic products [104]. With regard to protein synthesis, Bentley et al. (2008) [58] also noted that protein intake is necessary for muscle recovery and glycogen resynthesis.-Glycogen replenishment: It is critical to replenish muscle and liver glycogen in order to maintain performance in subsequent sessions [90]. Post-exercise CHO ingestion accelerates this process [95]. Frentsos and Baer (1997) [53] reported that dietary strategies combining increased carbohydrate and protein intake can improve recovery rates. Jeukendrup et al. (2005) [55] recommended 1.0–1.2 g/kg CHO per hour for post-exercise recovery to optimise glycogen synthesis.

It has also been observed that recovery can be influenced [76] by fibroblast growth factor 21 (FGF-21), a hormone which regulates lipid and CHO metabolism. This hormone promotes glucose uptake in muscle and fat cells by increasing CHO uptake and stimulating the production of brain-derived neurotrophic factor (BDNF), which regulates appetite and BM. These researchers studied these parameters following a study with seawater.

-Restoring Fluid and Electrolyte Balance: The loss of fluids and electrolytes during exercise can affect neuromuscular and metabolic function [105]. Adequate hydration can help to prevent cramps and premature fatigue [106]. Jeukendrup et al. (2005) [55] have already suggested that drinks containing sodium (Na) at concentrations of 30–50 mmol/L facilitate water absorption and prevent hyponatraemia. Bentley et al. (2008) [58] pointed out that maintaining fluid balance before and after exercise is important for delaying fatigue and improving recovery. Aragón-Vela et al. (2024) [84] conducted a study on hydration using mineral-rich deep seawater (DSW) and found that DSW could help maintain muscle strength during isometric exercise.-Regulation of the immune system: Intense exercise can temporarily suppress immune function, thereby increasing the risk of infection [107]. However, recovery promotes immune normalisation [108].

Among the factors influencing recovery, several factors are modulators of the efficiency of the recovery process, which is why it is so important to consider them all:-Training intensity and volume: Excessive training without adequate recovery can lead to overtraining syndrome, chronic fatigue and reduced performance [109].-Sleep: Deep sleep is essential for the release of anabolic hormones such as growth hormone, which promotes tissue repair [110]. Getting enough sleep is vital for muscle recovery and preventing injury [111].

Both of these factors have subsequently been addressed and combined by Hotfiel et al. (2019) [69].

-Nutrition: A balanced diet containing sufficient calories, protein, CHO and micronutrients is essential for facilitating repair processes [112].-Active vs. passive recovery strategies: Gentle exercise can improve circulation and speed up the removal of metabolites [113]. However, passive methods are also available, such as complete rest, massage or cryotherapy [114].

The following strategies are based on scientific evidence. They are for optimising recovery in triathletes:-Post-exercise nutrition: It is recommended that you consume a combination of carbohydrates and proteins within the first few hours after exercise [115]. For example, a 3:1 or 4:1 ratio of CHO to protein can increase protein synthesis and replenish glycogen stores [116]. Faster muscle recovery can also be achieved by taking 3 g/day of citrulline (CIT) and 300 mg/day of nitrate-rich beetroot extract (BRG), which may promote faster recovery [73]. However, this does not prevent exercise-induced muscle damage (EIMD) [117].-Hydration: Maintaining electrolyte balance involves drinking fluids with electrolytes according to estimated losses [118].-Recovery techniques: Sports massage can reduce muscle soreness and improve circulation [119]; cryotherapy can reduce inflammation [120]; and gentle stretching can help maintain flexibility [121]. Post-exercise compression appears to reduce muscle damage (Hotfiel et al., 2019 [69]), while cold water immersion is effective in reducing delayed-onset muscle soreness (DOMS) and inflammation; however, it may interfere with muscle glycogen recovery [69].-Adequate sleep: Prioritising a good night′s sleep is key. Some research suggests that sleeping for 7–9 h promotes the anabolic processes necessary for recovery [122].-Inadequate recovery can result in:-Overtraining: A condition characterised by persistent fatigue, reduced performance, and hormonal imbalances [123].-Overuse injuries: Tendinitis, stress fractures and muscle injuries often occur when proper tissue repair is not permitted [124].-Impaired immune system: Increased susceptibility to respiratory infections and other illnesses [125].

With this in mind, triathletes should remember that their training usually involves long sessions and frequent competitions over extended periods [3]. This requires personalised strategies that take this into account:-Training should be periodised to include loading and unloading phases [126]. Physiological status should be monitored using biomarkers such as hormone levels or resting heart rate [127].-Providing psychological support to help people cope with stress caused by multiple events [128].

### 4.3. Ergogenic Supplements: Scientific Evidence

Ergogenic supplementation is the use of substances or products. These are designed to improve athletic performance. They do so through specific physiological mechanisms [129].

Triathlon requires a unique combination of aerobic endurance, muscular strength, and the ability to recover quickly [130]. Muscle fatigue, glycogen depletion, dehydration and electrolyte imbalances commonly limit performance during competition [131]. The aim of ergogenic supplementation is to optimise these aspects in order to improve performance and reduce the risk of injury or medical complications [132].

The most commonly used ergogenic supplements in triathlons are:*a.* *CHO*

Consuming carbohydrates before, during and after exercise has been shown to effectively maintain blood glucose levels, delay fatigue and speed up recover [94]. The most common forms are gels, isotonic drinks and bars.

Frentsos and Baer, 1997 [53] indicated that increasing CHO intake delays fatigue and improves endurance performance. This is consistent with Jeukendrup et al., 2005 [55] findings on endurance performance. These findings were reaffirmed by Gillum et al., 2006 [56], who stated that CHO intake is essential for triathletes, and by Barrero et al., 2015 [63], who concluded that CHO supplementation is key to performance.

Tan et al., 2022 [77] reaffirmed this suggestion, adding that, in addition to improved performance, athlete health also improved. The method of ingestion appears to be an important factor, as Sareban et al., 2016 [66] demonstrated that carbohydrate ingestion in gel form did not enhance performance compared to liquid carbohydrate, but increased gastrointestinal discomfort. In this regard, Martínez-Olcina et al., 2022 [75] positively associated CHO (gel) intake with mood and mental health.

This variable is important in enabling athletes to comply with established recommendations. Cox et al., 2010 [61] observed that elite triathletes tended to meet pre-competition carbohydrate intake recommendations, but struggled to do so during the race. This suggests that palatability and prior training in intake are essential for implementation during competition [133].

McKay et al., 2020 [71] indicated that the periodisation of CHO intake is a useful tool for optimising adaptations to training and may increase reliance on fat as an energy source. Bennett et al., 2023 [79] reported that periodisation of CHO intake during sleep and light exercise improves metabolic adaptations and performance.

In addition, consideration should be given to whether CHO consumption aligns with the recommended glucose–fructose ratio, as this combination increases the oxidation of exogenous carbohydrates during exercise and minimises GI discomfort [64]. Appropriate dietary practices should also be followed, as studies indicate that inappropriate post-exercise dietary practices occur depending on age group, with less CHO being consumed than recommended [65].

*b.* 
*Electrolytes*


Excessive loss of sodium, potassium, magnesium and other electrolytes can lead to muscle cramps and alterations in neuromuscular function, which can affect performance in the various disciplines of triathlon [134]. Taking mineral salt supplements regularly helps maintain electrolyte balance and prevent these problems, as shown by McKay et al., 2020 [71] with Fe.

Several studies [135,136] have shown that electrolyte supplementation can improve performance during prolonged exercise by maintaining electrochemical balance and preventing complications related to dehydration or hyponatraemia. Sports organisations recommend electrolyte supplementation before, during and after exercise to optimise recovery and performance [137].

During a triathlon, particularly a long event, electrolyte loss (sodium, potassium, magnesium and calcium) through sweat can cause imbalances that affect muscle and nerve function [106]. Adequate electrolyte replacement helps to reduce the incidence of cramps during competition [106]. Studies [118,131] suggest that adequate electrolyte intake can maintain muscle and neuromuscular function, delay fatigue, improve endurance and help maintain thermal homeostasis during prolonged exercise.

Electrolyte supplementation is recommended for athletes participating in long-distance events, as it helps to prevent imbalances that could impact performance [137]. Evidence supports the strategic use of these supplements in triathlons to optimise performance and reduce the associated risks.

*c.* 
*Proteins and amino acids*


Adequate protein intake is essential for repairing the muscle damage caused by intense and prolonged triathlon training [138]. Essential amino acids, especially leucine, stimulate muscle protein synthesis via the mTOR pathway, thereby accelerating recovery [139].

Although their primary function is to aid muscle recovery after exercise, some studies suggest that branched-chain amino acids (BCAAs) may alleviate central fatigue during prolonged exercise [140].

Hotfiel et al., 2019 [69] have demonstrated that protein and amino acid supplementation promotes muscle repair and glycogen resynthesis, thereby establishing the ideal post-exercise dietary regimen for protein consumption [65]. During periods of intense exercise or a calorie deficit, protein protects muscle mass, and increased amino acid availability in the blood reduces protein breakdown [141].

Consuming protein after exercise can reduce delayed onset muscle soreness (DOMS). The rapid repair of damaged muscle tissue reduces inflammation and pain [142]. Adequate protein intake helps to maintain or increase lean body mass during periods of intense exercise [143]. Adequate protein intake enhances protein synthesis, contributing to improved aerobic performance [144].

Some studies suggest that taking branched-chain amino acid (BCAA) supplements may reduce central fatigue and improve endurance [145,146]. BCAAs (leucine, isoleucine and valine) play a role in energy production during prolonged exercise and may reduce the perception of fatigue [147]. Certain amino acids, such as beta-alanine, can increase muscle carnosine levels at certain concentrations, helping to buffer lactic acid and delay fatigue [148]. BCAAs can reduce the perception of central and peripheral fatigue by reducing ammonia accumulation and improving muscle protein synthesis [149]. Thus, amino acid supplementation can accelerate muscle repair after exercise, allowing for faster recovery [102]. Essential amino acids are also essential for maintaining and building muscle mass during periods of intense exercise. Some also have antioxidant properties that help to reduce the oxidative stress induced by prolonged exercise [150].

The scientific recommendations for triathletes can be summarized as follows:Daily intake of 1.2–2.0 g/kg body weight of protein, adjusted for training volume and intensity [143,151].Taking essential amino acids or BCAAs before or after exercise enhances recovery [152].Distribute protein evenly over several meals to maximise protein synthesis [153].

*d.* 
*Creatine*


Although it is traditionally associated with explosive power sports, some studies suggest that it may also be beneficial for endurance activities, as it can improve anaerobic capacity and recovery between sessions [154]. However, its use in triathlons is not as widespread as in other sports.

Creatine helps to rapidly replenish ATP (adenosine triphosphate), the main source of energy for short, intense activities [155]. This can be beneficial in certain triathlon segments, such as open water swimming, sprint cycling and fast transitions. Creatine can also promote gains in muscle mass, contributing to greater strength and endurance during the cycling and running phases. Some studies [154,156] suggest that creatine may reduce muscle damage and improve recovery after intense exercise, enabling sustained high performance throughout the competition. Creatine increases intracellular water content, which may positively affect muscle function and resistance to fatigue [157]. Although traditionally associated with anaerobic exercise, some studies suggest that creatine may indirectly benefit aerobic performance by improving recovery and reducing muscle fatigue [154].

Specifically for triathletes, creatine is not typically the main supplement used to improve performance in long events, as its primary effect relates to short, intense efforts [154]. It is recommended that a loading phase (20 g per day, divided into four doses over five to seven days) is followed by a maintenance phase (three to five g per day), although some studies suggest that it can be used without a loading phase [158]. Adequate hydration is essential to avoid possible side effects related to water retention [159].

*e.* 
*Caffeine*


It is one of the most widely researched ergogenic supplements. It acts as a central nervous system stimulant, increasing alertness and reducing the perception of exertion, thereby improving endurance performance [160]. However, its use should be monitored for potential adverse effects, such as insomnia or tachycardia.

Caffeine has been shown to enhance endurance in prolonged activities such as triathlons, primarily by reducing perceived fatigue and boosting motivation [161]. It can promote lipolysis, favouring the use of fat as an energy source, while preserving muscle glycogen stores, which is beneficial during prolonged exertion [162]. As a central nervous system stimulant, caffeine improves attention, concentration, and perception of effort during competition [163]. Studies suggest that moderate doses (3–6 mg/kg body weight), taken approximately 60 min before exercise, can significantly improve performance in endurance tests and combined events, such as triathlons [164]. Caffeine reduces the subjective feeling of fatigue, enabling you to maintain a higher pace for longer [164].

Despite these benefits, there are some important considerations:Reactions to caffeine can vary from person to person [165].It is advisable to carry out preliminary tests to determine personal tolerance [166].Excessive consumption can cause adverse effects such as insomnia, nervousness, and gastrointestinal problems [160].

The recommendation made by Wilson et al., 2015 [64] is for relatively low doses of caffeine, of approximately 3 mg/kg/1 h, to be taken just before training or competition.

*f.* 
*Beta-alanine*


This amino acid, a precursor of carnosine, can help to buffer the acids produced during intense exercise, thereby delaying muscle fatigue [167]. Reduced fatigue enables increased training volume and intensity, which can lead to improvements in overall fitness and competitive performance [168]. Studies show that beta-alanine supplementation can improve performance in activities involving repeated or sustained high-intensity efforts, such as triathlons (swimming, cycling and running) [148,169]. Furthermore, some studies suggest that beta-alanine may reduce the perception of fatigue and improve recovery between intense exercise sessions [170].

Most studies [148,171] suggest that daily doses of around 4–6 g, taken several times a day for at least 2–4 weeks, are necessary to optimise muscle carnosine levels and observe ergogenic effects.

Some people experience paraesthesia (a tingling sensation) as a side effect, which can be alleviated by taking smaller, more frequent doses [148]. The effects may vary between individuals, and further research is needed to understand the effect on triathletes.

*g.* 
*Nitrates*


Nitrates, which are commonly found in foods such as beetroot and other green leafy vegetables, have been studied for their potential ergogenic effects on athletic performance [172], including in triathlons.

The main scientific evidence for these effects is as follows:-Nitrates may reduce oxygen consumption during moderate- to high-intensity exercise, thereby improving muscle energy efficiency [172]. This is due to the conversion of nitrate to nitric oxide (NO), which promotes vasodilation and improves blood flow [173].-Some studies have shown that nitrate supplementation can increase aerobic capacity and delay fatigue, enabling athletes to maintain a higher pace for longer [174], which is crucial in triathlons.-Some evidence suggests that nitrates may reduce the subjective perception of exertion, thereby allowing for greater tolerance to prolonged exercise [175,176].-There is stronger evidence in endurance sports such as cycling and running, but studies also suggest potential benefits in combined events such as triathlons [177].-Most studies utilise nitrate doses ranging from 300 to 600 milligrams, typically administered two to three hours prior to physical exertion, with the aim of optimising the nitrate′s effects. Examples of sources of nitrate include beetroot juice [178].-Although nitrate supplementation is generally considered safe at the recommended doses, it is advisable to consult a professional beforehand, particularly if you have pre-existing medical conditions [174].

Gonçalves et al. (2011) [62] investigated the ergogenic benefits of organic grape juice (OGJ) in triathlons. They observed an increase in total plasma polyphenol concentration after 20 days of OGJ consumption. OGJ was found to improve antioxidant capacity, glucose homeostasis, and microcirculatory parameters in endurance athletes. Consistent with other studies, the consumption of grape juice is suggested to have a positive effect on the health of athletes and reduce the risk of cardiovascular disease.

Burgos et al. (2022) [73] investigated the impact of long-term supplementation with 3 g/day of citrulline and 2.1 g/day of nitrate-rich beetroot extract on maximal strength, endurance, power, and aerobic performance in male triathletes who were already trained. Supplementation with CIT+BRG was found to significantly improve muscle mass and mesomorphy, indicating an improvement in body composition. It also improved lower-body-specific strength (e.g., jumping ability) and core strength, as well as improving Cooper test scores. In a related study, the same authors found that, although CIT+BRG did not improve markers of exercise-induced muscle damage (EIMD), it prevented an increase in cortisol and a decrease in the testosterone/cortisol (T/C) ratio. Furthermore, this combination promoted greater distance covered in the Cooper test after nine weeks of supplementation.

*h.* 
*Taurine*


Taurine is an amino acid whose potential ergogenic effects in the field of athletic performance have been studied [179]. While the scientific evidence remains inconclusive, some studies suggest that taurine may provide advantages in endurance sports such as the triathlon [67].

Taurine may stabilise cell membranes and reduce oxidative stress during intense exercise, thereby reducing muscle fatigue and improving endurance [180]. It plays a role in regulating calcium, potassium and sodium levels in muscle cells, thereby helping to maintain muscle function and prevent cramps during prolonged exercise [181]. Its antioxidant properties can reduce the oxidative damage caused by intense exercise, contributing to faster recovery and better adaptation to training [182]. Some studies suggest that taurine may improve cardiac and vascular function, enabling greater oxygen delivery to the muscles during prolonged exercise [180,183]. Taurine may also influence energy metabolism, promoting the more efficient use of glycogen and fat reserves during endurance exercise [183].

Some clinical trials have reported improvements in parameters such as aerobic capacity, reduced perceived exertion, and reduced markers of muscle damage following taurine supplementation [184]. The recommended dose is 1–6 g per day, typically taken before or during training or competition [151].

De Carvalho et al. (2017) [67] found that taurine supplementation increased plasma taurine levels and decreased markers of oxidative stress in triathletes. This suggests that taurine may protect against oxidative stress, although it did not improve aerobic parameters.

*i.* 
*Probiotics*


The ergogenic effects of probiotics on triathlon performance and athletic performance in general have been the subject of research in recent years [185]. While the evidence is still emerging, some scientific studies suggest that probiotics may indirectly improve performance through various mechanisms [186,187]. Endurance athletes, such as triathletes, are susceptible to upper respiratory tract infections (URTIs), particularly during intense training and competition periods [15]. Probiotics can modulate the gut microbiota, thereby boosting immune response and reducing the incidence and duration of these infections [188]. This enables a more consistent training volume and prevents losses due to illness.

Some probiotics have anti-inflammatory and antioxidant properties, while help to reduce the oxidative stress caused by intense exercise [189]. Reducing inflammation can speed up recovery and improve adaptation to exercise. Gastrointestinal problems are common in endurance athletes and can affect test performance. Probiotics can help to maintain the integrity of the intestinal mucosa, reduce symptoms such as bloating, diarrhoea and nausea, and optimise the absorption of essential nutrients for performance [190].

Some studies suggest that probiotics may improve metabolic efficiency by influencing carbohydrate and fat metabolism in the gut microbiota, thereby facilitating the more efficient use of energy reserves during prolonged exercise [191]. In this regard, emerging evidence suggests that certain probiotics may positively influence mood and reduce feelings of fatigue, possibly via the gut–brain axis [192]. This could result in greater motivation and mental stamina during competition.

Most studies have been conducted on the general population or endurance athletes in general. There is less specific evidence relating to triathletes. Huang et al. (2019) [70] investigated the effects of Lactobacillus plantarum PS128 supplementation on potential physiological adaptations in triathletes. *L. plantarum* PS128 improves physical performance by modulating inflammation, oxidation and metabolism.

*j.* 
*Seawater*


Seawater contains a balanced concentration of electrolytes, such as sodium, magnesium, calcium and potassium, which are essential for maintaining muscle and nerve function during prolonged exercise. Some studies suggest that seawater may help to prevent muscle cramps in endurance athletes [193].

The presence of salts in seawater may improve the body′s absorption and retention of water, helping to maintain adequate hydration during prolonged activities such as triathlons [76].

Some scientific studies suggest that certain minerals found in seawater, such as magnesium, can reduce muscle fatigue and improve recovery after exercise [193].

It has been suggested that the trace elements and bioactive compounds found in seawater could help to reduce oxidative stress and inflammation following intense exercise. However, further conclusive research in this area is required [194]. While preliminary studies indicate potential ergogenic benefits, hard scientific evidence remains limited. Most research has focused on supplements derived from seawater or controlled saline solutions.

Acevedo et al., 2022 [76] demonstrated that the use of seawater as an ergogenic supplement did not directly enhance performance or diminish perceived exertion. However, it helped maintain plasma volume, prevented rapid increases in haematocrit and positively modulated post-exercise inflammatory and metabolic responses by increasing IL-6 and apelin production.

It should be noted that the direct consumption of untreated seawater can be risky due to its high salt content and potential contamination. For this reason, some athletes opt for isotonic solutions or processed seawater supplements to ensure safety and efficacy. However, the scientific evidence is inconclusive, and many studies have been conducted under controlled conditions that differ from those experienced in a triathlon.

*k.* 
*Krill oil*


Krill oil is rich in omega-3 fatty acids, primarily EPA and DHA, which have anti-inflammatory properties. It can therefore help to reduce muscle inflammation and speed up recovery after intense training or competition [195].

Omega-3s are essential for improving cardiovascular health, which helps to maintain efficient blood flow and adequate muscle oxygenation over long periods of time—a key factor in endurance sports [196].

Some studies suggest that taking krill oil may reduce muscle soreness and fatigue after exercise, thereby enhancing training and competition capacity [197]. The astaxanthin found in krill oil, for example, has antioxidant properties that may protect muscle cells from the oxidative stress generated during intense exercise [198].

Although the evidence is still limited, some studies suggest that omega-3 fatty acids may positively affect energy metabolism by promoting the more efficient use of fat as an energy source during prolonged exercise [196].

Most of the available scientific evidence is preliminary or has been conducted in animal models or small groups of people. There is limited specific clinical research in triathletes, so conclusions should be interpreted with caution. However, the most consistent and relevant effects for endurance athletes appear to be the anti-inflammatory and antioxidant benefits.

In this context, Storsve et al. (2020) [72] used krill oil supplements to maintain higher levels of choline and dimethylglycine, particularly following long-distance events. A significant decrease in serum choline concentrations was observed during the competition, with a greater decrease seen in long-distance events. These findings suggest that krill oil supplements may help to maintain choline levels and improve recovery during endurance events [72].

Limited training in sports nutrition, conflicting online information, excessive focus on physical training and inadequate adaptation of diets to individuals are all factors that limit nutritional knowledge in sports. Consequently, scientific evidence concerning the efficacy and safety of supplements is imperative. Numerous studies have shown that certain supplementation strategies can improve performance in prolonged tests similar to triathlons, with some even suggesting that these strategies can enhance performance in events lasting up to eight hours:I.Carbohydrates: Consuming them during prolonged exercise significantly increases the time it takes to become fatigued [199].II.Electrolytes: Adequate replenishment prevents cramps and maintains neuromuscular function [134].III.Caffeine: Significant improvements in race times were observed when caffeine was administered in moderate doses [200].

However, it is crucial to adhere to the recommended dosage to avoid adverse effects. The practice of supplementation should be based on robust scientific evidence. Preliminary tests are recommended during training to assess tolerance. If triathletes do not follow the recommended nutritional guidelines, the following points may be taken into account:Decreased performance: A deficiency in carbohydrates can lead to a reduction in muscle and liver glycogen stores, which can subsequently affect endurance during the event.Lack of protein or micronutrients has been shown to lead to slower muscle recovery and an increased chance of injury.There is a risk of dehydration or hyponatraemia. Poor hydration management has been shown to increase the risk of dangerous complications.GI issues: It has been demonstrated that ingesting specific foods or supplements in quantities that exceed recommended levels can result in adverse effects during competitive events.The long-term health implications of the following factors must be given due consideration: It is important to note that malnutrition and deficiencies can affect athletes’ overall well-being.

It is imperative to prioritise a balanced diet over reliance on supplements, as a balanced diet provides all the nutrients that your body needs, whereas supplements are not necessarily necessary, as they do not provide all the nutrients that your body needs.

## 5. Strengths, Limitations, Future Research Lines, Practical Applications

This timely review incorporates recent studies to provide an up-to-date perspective, highlighting under-researched areas to inform future research. Scientific evidence suggests that dietary strategies and supplements, such as omega-3 fatty acids (OGJ) and branched-chain amino acids (BCAA), can improve athletic performance. Nutritional strategies should be tailored to the individual based on factors such as age, gender, sport and dietary preferences [101]. When implemented correctly, these strategies can lead to significant improvements in physical performance, endurance and recovery [23]. Adequate nutrition has been shown to contribute to better overall health and injury prevention by strengthening the immune system and promoting muscle recovery [201]. Increased awareness of the importance of nutrition in sport has led to greater education and training for athletes and coaches.

The results of dietary interventions can vary greatly from person to person, which makes it difficult to make universal recommendations. The lack of standards for the dosage and timing of ergogenic supplements can cause confusion for athletes and coaches [202]. Additionally, some supplements may be expensive or unavailable to all athletes, limiting accessibility. Misleading sports nutrition information can lead athletes to make poor decisions, raising regulatory and ethical concerns about certain ergogenic strategies [203]. A potential limitation of such a review is that the validity and generalisability of the results may be affected by the studies included. The limitations of the present study are as follows: small sample size; possible biases in participant selection; differences in methods; lack of control for confounding variables; and variable methodological quality among studies. These limitations must be acknowledged when interpreting the conclusions and determining the applicability of the findings in different contexts.

In the future, it would be interesting to investigate how different supplements interact with each other, and the combined effect of this interaction on athletic performance. Therefore, longitudinal studies should be conducted to assess the long-term effects of different nutritional strategies on athletes’ performance and overall health [204]. Personalised nutrition should be developed to tailor dietary and supplementation approaches based on factors such as genetics and gut microbiota [205]. Furthermore, it would be valuable to investigate how nutritional strategies affect physical performance as well as psychological aspects such as motivation and self-confidence. This could inform best practice for active recovery compared to passive methods when combined with nutrition.

Practical applications should focus on creating personalised nutrition plans for specific sports, including the optimal timing and use of ergogenic supplements. It is crucial to provide up-to-date, evidence-based sports nutrition education through workshops and resources to enable informed decision-making. Technological tools can track food intake, sleep, and recovery, enabling the dynamic adjustment of nutritional strategies in conjunction with training programmes to maximise performance and recovery [206].

## 6. Conclusions

It is well-documented that triathletes often do not adhere to the recommended nutritional guidelines for CHO and protein, particularly during the critical post-exercise recovery period. This deficiency can compromise muscle glycogen replenishment and the repair of damaged muscle tissue, ultimately exerting a negative effect on future performance. While scientific research has demonstrated the efficacy of certain ergogenic aids—such as caffeine, which is known for its ability to reduce perceived exertion and improve endurance; beetroot extract, which promotes vasodilation and improves oxygen use efficiency; and probiotics, which modulate the gut microbiota and potentially improve nutrient absorption and immune function—it is important to recognise that the evidence regarding their long-term effects, as well as their potential interactions with each other and with the usual diet, is incomplete and requires more thorough and rigorous research.

In addition to the use of nutritional supplements, it is crucial to emphasise the important role of improved and more accessible nutrition education for triathletes. This education should go beyond mere information transmission to focus on cultivating practical competencies such as meal planning, nutrition label interpretation and assessing individual requirements. Alongside education, implementing personalised dietary strategies designed specifically for each athlete′s metabolic and physiological demands is crucial for optimising endurance performance. This personalisation process should consider factors such as training volume and intensity, body weight and composition, dietary preferences, and potential intolerances or allergies. Finally, it is crucial to mitigate the risk of Relative Energy Deficiency in Sport (RED-S), a serious condition with negative short- and long-term health consequences including hormonal disorders, bone problems, altered immune function, and psychological issues. To prevent RED-S, the athlete′s energy status must be regularly monitored, and a holistic approach must be adopted that gives equal priority to athletic performance and overall well-being.

## Figures and Tables

**Figure 1 nutrients-17-01846-f001:**
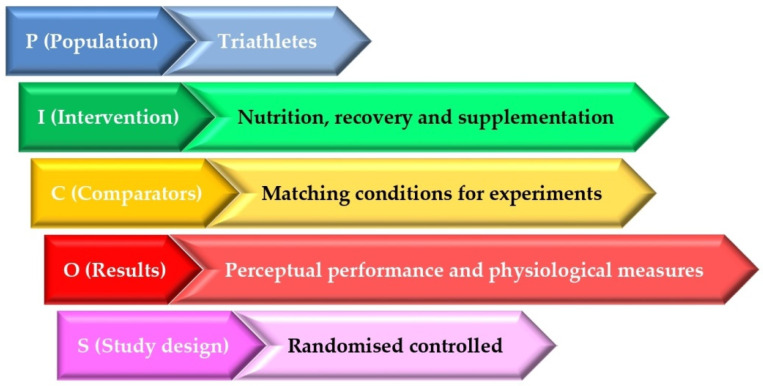
PICOS model.

**Figure 2 nutrients-17-01846-f002:**
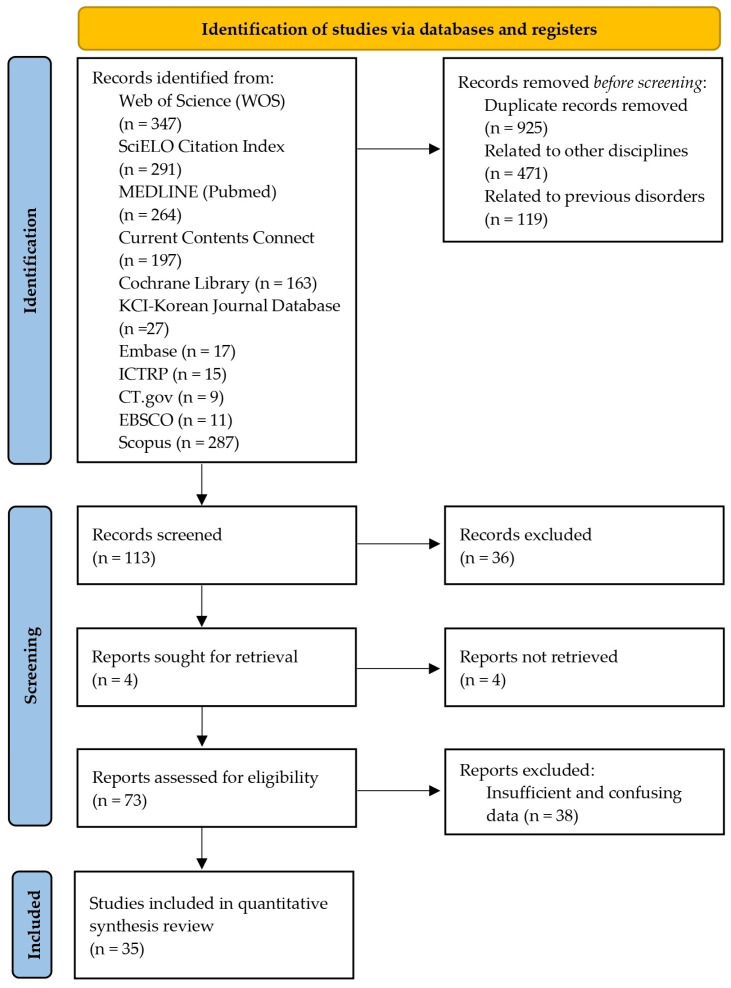
Study selection flowchart [47]. (This work is licensed under CC BY 4.0. To view a copy of this license, visit https://creativecommons.org/licenses/by/4.0/ (accessed on 30 October 2024).

**Figure 3 nutrients-17-01846-f003:**
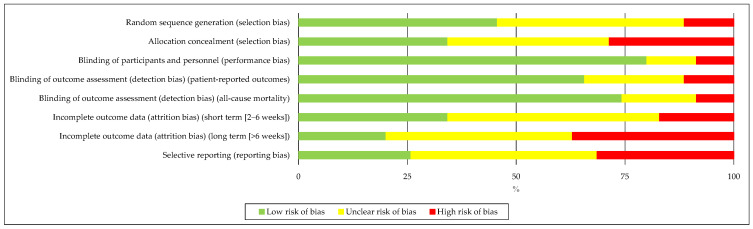
Bias risk graph. The colours used to indicate risk are as follows: green for low risk, yellow for uncertain risk, and red for high risk. This graph shows the overall risk of bias in each group. For example, the size of the green rectangle shows how many studies have a minimal risk of bias.

**Figure 4 nutrients-17-01846-f004:**
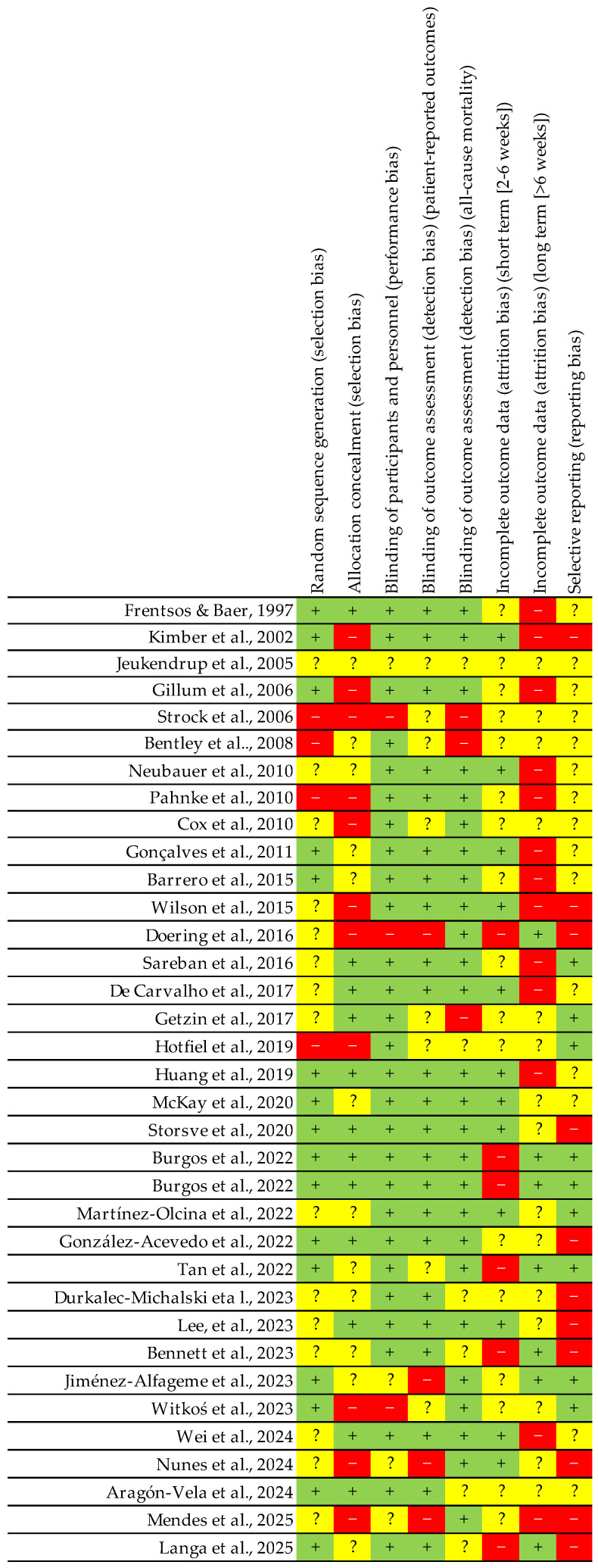
Summary of risk of bias, indicating the level of bias for each domain in each study. Red colour: elevated level of bias; Green colour: low level of bias; Yellow colour: unclear level of bias; +: possibility of low bias level; −: possibility of high bias level; ?: possibility of undetermined bias [16,35,53,54,55,56,57,58,59,60,61,62,63,64,65,66,67,68,69,70,71,72,73,74,75,76,77,78,79,80,81,82,83,84,85].

**Figure 5 nutrients-17-01846-f005:**
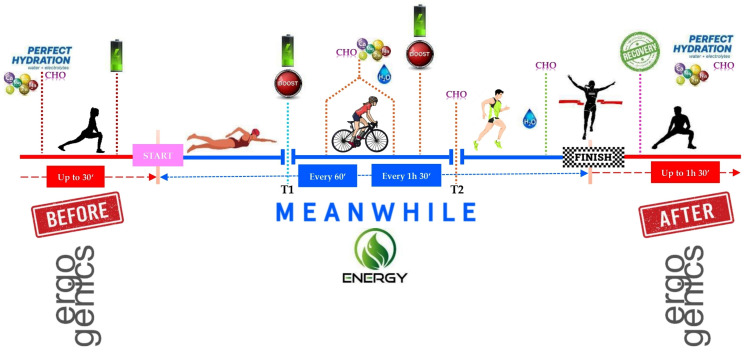
Potential nutrition strategy.

**Table 1 nutrients-17-01846-t001:** Evidence levels of selected studies using the Oxford Quality Rating System [50].

Research	Level
Frentsos and Baer, 1997 [53]	B
Kimber et al., 2002 [54]	A
Jeukendrup et al., 2005 [55]	A
Gillum et al., 2006 [56]	B
Strock et al., 2006 [57]	A
Bentley et al., 2008 [58]	A
Neubauer et al., 2010 [59]	A
Pahnke et al., 2010 [60]	A
Cox et al., 2010 [61]	A
Gonçalves et al., 2011 [62]	B
Barrero et al., 2015 [63]	A
Wilson et al., 2015 [64]	B
Doering et al., 2016 [65]	B
Sareban et al., 2016 [66]	A
De Carvalho et al., 2017 [67]	A
Getzin et al., 2017 [68]	A
Hotfiel et al., 2019 [69]	A
Huang et al., 2019 [70]	A
McKay et al., 2020 [71]	A
Storsve et al., 2020 [72]	A
Burgos et al., 2022 [73]	A
Burgos et al., 2022 [74]	A
Martínez-Olcina et al., 2022 [75]	A
González-Acevedo et al., 2022 [76]	A
Tan et al., 2022 [77]	A
Durkalec-Michalski eta l., 2023 [78]	A
Lee, et al., 2023 [35]	A
Bennett et al., 2023 [79]	A
Jiménez-Alfageme et al., 2023 [80]	A
Witkoś et al., (2023) [81]	B
Wei et al., 2024 [82]	B
de Oliveira et al., 2024 [83]	B
Aragón-Vela et al., 2024 [84]	A
Mendes et al., 2025 [85]	B
Langa et al., 2025 [16]	A

**Table 2 nutrients-17-01846-t002:** Studies included in the work.

Knowledge Area	Journal	Q	Authors	Population	Age Ranges (Years)	Triathlon Type	Method	Intervention	Variables	Results Analysed	Main Conclusions
Md	Int. J. Sport Nutr. Exerc. Metab.	1	Frentsos and Baer, 1997 [53]	Elite 4 ♂/2 ♀	31.0 ± 3.0	IM	DB	Modification of Int of N; + of Int of CHO; adjustment of Pt for post-exercise; regulation of Int of fats; plans for greater H	Total calories consumed; proportion of mcN scheduled for the test; moment of Int of the N	Composition of nutritional supplements; dietary Int; food Int patterns; competition dietary practices;	An adequate Int of N is necessary to improve the P of End
											The + of Int of CHO allows you to delay the F and improve the End
											The B between mcN favours Rcv
Md	Int. J. Sport Nutr. Exerc. Metab.	1	Kimber et al., 2002 [54]	10 ♂–y 8 ♀	35.25 ± 8.3	IM	DB	Laboratory test; incremental ramp test; MD&W [86]	hg; BW; %BF; FM; FFM; BMI; EI; EE; VO_2_; s; CHO intake	EI during cycling and running; mcN Int and Na; EE; EB	Despite the negative EB, the average EI seemed sufficient before reaching substrate depletion
											CHO Int adequate to satisfy the maximum rates of plasma glucose oxidation by skeletal muscle
											The increase in EI may have improved the performance of the ♂
											The increase in total CHO and EI were not related to faster completion times in ♀
SM	Sport. Med.	1	Jeukendrup et al., 2005 [55]	-	-	-	DB	Ig strategies of CHO, H and Rcv	Int of CHO; H; synthesis of G; sporting P; medical problems	Loading of CHO; H; Post-exercise recovery	Nt is important for improving P and Rcv
											Int of CHO and an H with Na help to maintain the E and prevent the Hn
											A well-designed nutritional strategy optimises P and reduces health risks
SM	Int. J. Sports Physiol. Perform.	2	Gillum et al., 2006 [56]	1 ♂	38	LD	DB	Analysis of muscle G depletion and resynthesis before, during and after competition	Consumption of CHO, Pt and fats, energy excretion and oxidation of substrates, muscle G levels	BW; muscle G; CHO consumption	CHO intake is essential
											Eccentric muscle damage from running can affect G resynthesis
STR	Phys. Med. Rehabil. Clin. N. Am.	2	Strock et al., 2006 [57]	Triathletes of diverse levels, from amateur to professional	-	-	S	Theoretical revision	Nt and H strategies; thermal problems and temperature regulation	Conditioning; nutritional aspects; thermoregulation;	Nutritional planning and adequate H are fundamental for the P and safety
STR	J. Sci. Med. Sport	1	Bentley et al., 2008 [58]	-	-	-	S	Analysis of strategies to improve P in running	EC; CHO; Pt; fluid loss; rest and recovery; V and I; thermal regulation; VO_2max_; lactate threshold	Adequate consumption of CHO pre- and post-test; Int of Pt post-exercise; H; sweating; electrolytes	Individual Nt, H and Rcv are needed to help the body readapt
											Acclimatisation is important to avoid F and improve heat tolerance
											Well-structured training is important (St, End, and s)
Md	Br. J. Nutr.	1	Neubauer et al., 2010 [59]	42 ♂	35.3 ± 7.0	IM	S	Free consumption during the competition	Endogenous and exogenous antioxidants in plasma; markers of oxidative stress; oxidative damage to DNA	Plasma antioxidant capacity; ORAC ratio and oxidative damage to DNA; carotenoids and α-tocopherol	The acute post-exercise antioxidant response of ultra-end can prevent oxidative damage to DNA
											Need for nutritional antioxidants during Rcv
											Special attention in the Rcv diet
STR	Med. Sci. Sports Exerc.	1	Pahnke et al., 2010 [60]	26 ♂–20 ♀	46.65 ± 10.65	IM	DB	[Na] in serum; BM; sweating; Int of Na and fluids	[Na] in serum and BW, pre/post-race; recording of fluids and [Na] during the race	Sweating analysis trial; Serum [Na]; race day results; Int during the race of mcN, Na, K, and fluid	The change in [Na] during the ultra-end exercise is related to the change in BMI and to the rate of Na loss through sweat
											The rate of Na loss through sweat and the changes in serum Na were clear in ♂, but not so clear in ♀
											Although the ♀ compensated their Na losses better through Na Int, the ♀ consumed more liquids than the ♂
											This higher fluid Int in the ♀ may partly explain their slightly greater decrease in serum Na on the day of the race than in the ♂
Md	Int. J. Sport Nutr. Exerc. Metab.	1	Cox et al., 2010 [61]	36 ♂–15 ♀ elite	Under-23	OT	S	Self-reported prerace and during-race nutrition data were collected at three separate OT events	CHO Int pre/post and during the race; race time; temperature and humidity during the race; differences in CHO Int according to gender	Nutrient-Int analysis	Elite triathletes usually comply with the Int recommendations of pre-race CHO, but have difficulties in achieving them during the race
Md	Clinics	2	Gonçalves et al., 2011 [62]	10 ♂	34 ± 7	-	DB	Measurement of blood parameters; capillary microcirculation	Glucose; insulin; uric acid; superoxide dismutase; functional capillary density; diameter of capillaries; resting erythrocyte flow velocity; time to reach peak velocity after occlusion; BW; %FM; %FFM; mcN	OGJ consumption for 20 days; HOMA2-IR index; uric acid; E-SOD; polyphenols; capillary density; peak erythrocyte flow velocity; time to reach peak s; diameter of afferent, efferent and apical capillaries; BW; %FM; %FFM	OGJ Int improved blood glucose, antioxidant capacity and microvascular function, without affecting BW or BC
											The observed benefits can be attributed to a high polyphenol content, which could favour P and cardiovascular health
FS	Nutrients	1	Barrero et al., 2015 [63]	11 ♂	36.8 ± 5.1	UE	S	Nutrition record and H; physiological variables	EI and EE during competition; BM; TBW; ICW; ECW; Int of CHO, proteins and lipids; relationship of fluid loss and Int of CHO, with sports P	P; mcN Int; Fluid and Na Int; EB; BM and bioimpedance bioelectricity variables; relationship between racing P and parameters assessed during the race	E demands significantly > than Int
											Loss of BM associated with fluid depletion and depletion of G and fat reserves
											CHO supplementation was key for P, and ∃ the need for personalised strategies to optimise Pt Int and lipids
ND	J. Am. Coll. Nutr.	2	Wilson et al., 2015 [64]	43 ♂–y 11 ♀	18–64	UE	S	CHO Int and its impact on digestion	CHO Int; proportion of glucose and fructose in food and drink; Na Int, proteins, fats and caffeine	Participant characteristics; food composition and saccharide Int; GI distress and associations with saccharide Int	CHO consumption does not fulfil glucose–fructose recommendations
											The higher consumption of glucose compared to fructose can lead to GI problems
Md	Int. J. Sport Nutr. Exerc. Metab.	2	Doering et al., 2016 [65]	101 ♂–81 ♀	41.5 ± 7.5	LD	S	Survey	Knowledge of recommended CHO and Pt post-exercise Int; actual consumption of CHO and Pt post-exercise; sources of information on Nt post-exercise; differences between age groups	Knowledge of post-exercise nutritional recommendations; post-exercise nutrition practices	Triathletes, regardless of age, have a poor understanding of the recommended post-exercise Nt
											M have inadequate post-exercise dietary practices, consuming less CHO and Pt than recommended
Md	Int. J. Sport Nutr. Exerc. Metab.	2	Sareban et al., 2016 [66]	9 triathletes	38.6 ± 10.7	LD	DB	CHO consumption during the test	Distance covered in the final race; RER; blood glucose and lactate; GI discomfort	Self-reported caloric Int during the 48-hr period before; P; fluid and CHO Int; glucose and lactate; RER; RPE; GI distress	The Int of CHO in gel does not improve the P in comparison with liquid CHO, but + GI discomfort
Ph	Front. Physiol.	1	De Carvalho et al., 2017 [67]	10 ♂	30.9 ± 1.3	LD	DB	Int Ta + low-fat chocolate milk	[Ta]; oxidative stress; protein metabolism; aerobic parameters	Oxidative stress and Pt metabolism marker levels; GSH; MDA; N ur; N_2_ B; vE; Urea; Creatinine	Supplementation with Ta did not improve Ae P, but it did improve oxidative stress and N_2_ balance, suggesting a possible protective effect on muscle catabolism
Md	Curr. Sports Med. Rep.	3	Getzin et al., 2017 [68]		-	-	DB	Int of CHO; special considerations for the obese triathlete	Distance; time; CHO Int differentiating between elite and non-elite triathletes and BM of each athlete	H; Int of CHO; caffeine; fluids; Na replacement; organic nitrates; exercise-induced GI syndrome; obesity	Reduction in the cost of O_2 in_ the whole body, especially in low to moderate I exercise, improvement in tolerance and P; improvement in cognitive functioning
											Decrease in blood pressure due to the effect of NO on vascular control
STR	Sports	1	Hotfiel et al., 2019 [69]	Olympic level	-	OT	DB	Compression therapy; CWI; active regeneration; nutritional supplementation; sleep quality	Use of garments and intermittent pressure devices; CWI; impact of sleep quality and quantity on Rcv; active regeneration; supplementation with Pt, BCAAs, omega-3; H	EIMD; post-exercise compression; CWI; sleep pattern; supplementation with Pt and BCAAs; active regeneration	CWI is effective for the reduction of DOMS and inflammation, but could affect muscle G Rcv
											A comprehensive approach that combines rest, nutrition and active Rcv optimises P
											Personalisation is key, as each athlete responds≠
											Post-exercise compression appears to be effective in reducing MD, although there is no consensus on its impact on P
											Muscle damage induced by EIMD affects P
											Sleep is fundamental for muscle Rcv and injury prevention
											Supplementation with Pt and BCAAs favours muscle repair and G resynthesis
											Active regeneration, such as foam rolling and light pedalling after competition, may aid Rcv, but evidence is limited
ND	Am. J. Clin. Nutr.	1	Huang et al., 2019 [70]	34 triathletes	20.93 ± 0.93	ST	DB	Daily supplementation with *Lactobacillus plantarum* PS128 capsules or placebo	CK, TRX, MPO), (TNF-α, IL-6, IL-10), (VO_2max_, Wingate), plasma amino acids; muscle F (lactate, ammonia), BC	BC Pre-Post SS; F and I-R BI; IC after IE; KI and MPO after IE; An and Ae ExC; Free AA Content after SS	Supplementation with *L. plantarum PS128* + post-exercise Rcv
											+ levels of essential amino acids and maintains P after intensive exercise
											Potential ergogenic aid for End athletes
Md	Int. J. Sport Nutr. Exerc. Metab.	1	McKay et al., 2020 [71]	4 ♂–7 ♀ elite	24.45 ± 2.5	-	DB	CHO Prdt; inflammation and immune function; HIGH; LOW	HIGH; LOW; HIT; LIT; physiological and metabolic biomarkers; VO_2max_; HR; Int mcN; state of health	Effects of CHO Prdt on Fe regulation; inflammation and immune function; response to training and energy metabolism	CHO Prdt can be a useful tool for optimising adaptations to training with adequate management of the Fe Int
											CHO PRT using the LOW strategy does not negatively affect inflammation or immune function
											The restriction of nocturnal CHO + the post-exercise hepcidin response
											Athletes in the LOW condition showed > dependence on fats as a source of E
											The P in LI sessions was not affected by the CHO restriction
ND	Front. Nutr.	1	Storsve et al., 2020 [72]	35 ♂–12 ♀ elite	40.45 ± 8.6	IM/OT	DB	Daily Spp before competition	Effect of krill oil supplementation on [choline] and its metabolites	Choline; betaine; DMG; other choline metabolites	Preventing [choline]—during long competitions, improving Rcv and P
FS	Nutrients	1	Burgos et al., 2022 [73]	32 ♂	32.17 ± 4.87	-	DB	Spp with Ci and/or BE rich in nitrates; physical tests	HJUMP; DYN; 1-MAT; Ct; Anthp	BC; somatotype; maximal St; End-St; Ae power	The combination of Ci and BE powers Ae
ABS	Biology (Basel)	1	Burgos et al., 2022 [74]	32 ♂	34.37 ± 7.08	-	DB	Supplementation	Dietary evaluation; hormones generated; Ct; Anthpc measurements	E and mcN Int; Anthp and BC; Ct; Serum EIMD markers; T-C status; T/C ratio	The combination of Ci-BE muscle damage markers
											T/C
											Rcv y la P
OG	Gels	2	Martínez-Olcina et al., 2022 [75]	10 ♂	26.0 ± 8.7	-	DB	Mouthwash	RPE; FS; FAS; POMS; blood glucose, sprints, and dietary habits	RPE; Fs; FAS; POMS; Int of CHO gel	Consumption of gel with CHO and mood
											RPE, activation, blood glucose levels and P in Sprint
											The level of activation over time
											Pleasant sensation with a lower proportion in terms of RPE
PHEOH	Int. J. Environ. Res. Public Health	2	González-Acevedo et al., 2022 [76]	10 ♂	38.8 ± 5.62	-	DB	Specific H	Haematological measurements; cytokine and cytokine Pt	Anthp and physiological characteristics; alterations in blood cytokine and myokine Pt levels	H, keeping plasma V constant
											Rcv time in End
											Use of fats as a source of E and metabolic adaptation
FS	Nutrients	1	Tan et al., 2022 [77]	12 ♂–9 ♀ elite	18.9 ± 1.6	-	S	Nutrition in training; H; mcN; micronutrients; supplements	SNKQ; dietary Int assessment	Sports nutrition knowledge; dietary Int; E; CHO; Pt; Fat; Ca; Fe	Nutritional knowledge and caloric intake to the demands of training
											+ Int of Ca and CHO, optimising P and health
FS	Nutrients	1	Durkalec-Michalski et al., 2023 [78]	18 ♂–2 ♀	32 ± 7	-	DB	An observational study carried out during two macrocycles of training specific to triathlon	Dietary record for three consecutive pre-test days; ICT to evaluate aerobic fitness; mineral content analysis	BM; h; BC; FM; FFM; TBW; EI; Pt; CHO; fat and dietary fibre Int; SFA; MUFA; PUFA; vitamins and minerals; Texh, VO_2max;_ VO_2VT_; %VO_2max_VT_); HR_max_; HR_VT_; W_VT;_ niveles de Cu, Fe, Zn, Ca y Mg	E and nutritional values of the usual diets between training and competition
											Fe content that may indicate an increase in nutritional needs during the competition period
											Ae capacity during the competition period
											Meaningful relationship between [Ca] and the absolute maximum uptake of O_2_
											% of maximum uptake of O_2_ in the VT
											Lack of dietary periodisation based on training macrocycles
FS	Nutrients	1	Lee et al., 2023 [35]	9 ♂–3 ♀	49.42 ± 5.9	L	DB	Double-blind, crossover, placebo-controlled laboratory trial	[GSH]; superoxide dismutase, catalase, biological antioxidant potential	MF; skeletal muscle oxygenation; CF; BS; RPE	vC with natural antioxidant is more effective for metabolic function, skeletal muscle oxygenation, cardiac function, and antioxidant function in an End activity than a single SS of vC
SM	Scand. J. Med. Sci. Sport.	1	Bennett et al., 2023 [79]	23 ♂	34 ± 7	L	DB	Randomised control trial; battery of physiological tests; registered dietary intake; F qualification; H	V̇O_2max_; training program LIT and HIT; dietary Int; submaximal cycling test; TT	Training response; submaximal cycling test; TT; dietary Int	CHO Prdt improves metabolic adaptation, and the P
											Prdt of CHO and HS did not provide additional benefits; in fact, it can impair the positive adaptations associated with the SL-TL diet
											HS can compromise the oxidation capacity of CHO, which calls into question the effectiveness of training with low availability of CHO in warm conditions
FS	Nutrients	1	Jiménez-Alfageme et al., 2023 [80]	165 ♂–67 ♀	34.79 ± 9.93	-	S	Validated SS consumption questionnaire	SS consumption; BMI; type of SS; reasons for consumption; source of recommendation; sex; competitive level	SS consumed; distribution SS on sex, type, competitive level	Consumption of SS is high among triathletes
											Professional nutritional advice is frequent
											Considerable use of SS with little scientific evidence
											20% had or had had monthly MD that could indicate an immediate risk of LEA
											10% of participants were at physiological risk and of P related to RED-S
											The number of reported injuries was higher in ♀ with MD, which may indicate that the lack of oestrogen can lead to a higher risk of injury
FS	Nutrients	1	Witkoś et al., 2023 [81]	30 ♀	33.5 ± 9.2	IM	DB	BC; LEAF-Q questionnaire	Incidence of injuries; GI problems; menstrual cycle disorders	hg; BM; BMI; FM; VAT; FFM; MM; TBW; ECW; monthly cycle disorders	LEAF-Q allowed for the early detection of FAT symptoms in several of the triathletes studied
											20% of the triathletes had or had had monthly cycle disorders that could indicate an immediate risk of LEA
											10% of the triathletes were at physiological and performance risk related to RED-S. The number of injuries reported in ♀ triathletes was higher in those ♀ with menstrual disorders compared to those ♀ without them, which may indicate that a lack of oestrogen can lead to a higher risk of injury
SM	Sport. Med. Heal. Sci.	2	Wei et al., 2024 [82]	13 elite triathletes	20.3 ± 0.5	OT/ST	S	Stroop colour–word test	Identify metabolically active regions; haemodynamic measurements	Regions with high glucose uptake; Stroop colour–word test; haemodynamic changes	Connection between the rectum and cognitive P
											Powerful intervention capable of improving cognitive P
SM	Med. Sci. Sports Exerc.	1	de Oliveira et al., 2024 [83]	724 ♂	38 ± 10		S	Questionnaire	Age, BMI, height, whether or not they take SS, nutritional advice, and the scope of this advice	Use of SS by categories; number of SS; amount of SS used; nutritionist guidance	The majority of triathletes took SS, 25% did not receive services from professionals
Md	J. Clin. Med.	1	Aragón-Vela et al., 2024 [84]	19 ♂	39.0 ± 4.25	-	DB	DSW supplementation; compared with two additional H conditions: isotonic placebo and tap water	H; Isometric muscle St; CMJ	Anthp; time; BM; RPE; and temperature of the test; CMJ; isometric muscle St test	Preserve muscle St in isometric exercises post-exercise
											CMJ
											Delaying muscle F and preserving post-exercise muscle function in End
Ph	Braz. J. Med. Biol. Res.	3	Mendes et al., 2025 [85]	61 ♂–11 ♀	34.5 ± 8.5	OT	S	Survey conducted using an online questionnaire on the Google Forms platform. The questionnaire included multiple choice and open questions.	Amount of training; gastrointestinal discomfort when ingesting CHO; types of CHO, whether they had nutritional advice and the time of the competition; gender; strategy of CHO use; amount of CHO	48.6% of triathletes reported having compensated for CHO before competitions. 86.1% of the triathletes had used CHO during competitions. Only two athletes ingested 60 g/h, and the average intake was 22.1 g/h	The quantities ingested were not always those recommended because only some of the participants received professional advice
FS	Nutrients	1	Langa et al., 2025 [16]	20 ♀	5.5 ± 2.5	IM	DB	Three-day food diary form; training diary; LEAF-Q	Baseline characteristics of high-P; dietary assessment; REE; EEE; PAL; TEE; LEA in females’ questionnaire	LEA in female’s questionnaire scores; diet; REE; aTEE; rTEE; aEI; rEI; aED; rED; LLEA; H-LEA; training load; dietary Int of mcN; PPROT; PUFA; PUFAn6; PUFAn3; MUFA; SFA; fibre Int; dietary Int of micronutrients—Fe and Ca	30% of the participants were at risk of suffering RED-S, and 50% presented at least one symptom related to LEA
											The diets consumed by triathletes who presented menstrual dysfunction, gastrointestinal symptoms and/or lesions were richer in fibre, differed in the type of Pt and in the proportion of fatty acids with respect to the diets of triathletes without LEA symptoms, who consumed more saturated fatty acids.
											Diets rich in fibre and vegetable Pt may put ♀ at risk of RED-S.
											The composition of dietary FM is essential for adequate E Int. Greater dependence on free fatty acids as E substrates

∃: there is; −: reduction; []: concentration; +: increase; <: lesser; >: greater; ≠: differences; ±: similar; 1-MAT: 1-Min Abdominal Test; AA: amino acid; ABS: agriculture and biological sciences; aED: absolute energy deficit; aEI: absolute energy intake; Ae: Aerobic; AIS: Australian Institute of Sport; An: anaerobic; Anthp: anthropometry; aTEE; absolute total energy expenditure; B: balance; BC: body composition; BCAAs: branched-chain amino acids; BE: beet extract; BF: body fat; BI: biochemical indices; BM: body mass; BMI: body mass index; BW: body weight; BP: blood pressure; C: cortisol; Ca: calcium; CCI: intraclass correlation coefficient; CFU: colony-forming units; CHO: carbohydrates; Ci: citrulline; CK: creatine kinase; CMJ: countermovement jump; Ct: Cooper test; CWI: Cold Water Immersion Therapy; Cu: Copper; DB: data base; DOMS: delayed onset muscle soreness; DMG: dimethylglycine; DSW: deep-sea water; DYN: Handgrip Dynamometer Test; E: energy; EB: energy balance; EC: energy consumption; ECW: extracellular water; EDM: endocrinology, diabetes and metabolism; EE: energy expenditure; EEE: Exercise Energy Expenditure; RER: respiratory exchange ratio; EI: Energy intake; EIMD: exercise-induced muscle damage; End: endurance; E-SOD: superoxide dismutase activity in erythrocytes; ExC: exercise capacities; F: fatigue; FAT: Female Athlete Triad; Fe: iron; FAS: Felt Arousal Scale; FS: Feeling Scale; FFM: fat free mass; FM: fat mass; FS: food science; Fs: sensation scale; G: glycogen; GI: gastrointestinal; GSH: glutathione; Green: favourable effect; hg: height; h: hour; H: hydration; Hb: Haemoglobin; HIGH: High CHO availability; HJUMP: Horizontal Jump Test; H-LEA: high LEA; HS: heat stress; HI: high intensity; HIT: high-intensity training; Hn: hyponatraemia; HOMA2-IR: homeostasis model assessment for the insulin resistance; HR: heart rate; I: intensity; ICW: intracellular water; ICT: incremental cycling test; IC: inflammation cytokines; IE: intense exercise; IL-10: cytokines; IL-6: cytokines; IM: Iron Man; Int: intake; IR: injury-related; KI: kidney injury; L: laboratory; Lat: lactate; LD: long distance; LI: low intensity; LIT: low-intensity training; LEA: Low energy availability; LEAF-Q: low energy availability in females questionnaire; LLEA: low LEA; LOW: low CHO availability; M: master; Md: medicine; Mb: Myoglobin; MD: menstrual disorders; MDA: malondialdehyde; Mg: magnesium; mcN: macronutrients; MD&W: method of Durnin and Womersley; MD: muscle damage; MDA: malondialdehyde; Mg: magnesium; MgO: magnesium oxide; MM: muscle mass; MPO: myeloperoxidase; MUFA: monounsaturated fatty acids; N: nutrients; Nt: nutrition; N_2_: nitrogen; Na: sodium; ND: nutrition and dietetics; NO: Nitric oxide; Nur: 24 h nitrogen excretion O_2_: oxygen; OG: organic chemistry; OGJ: organic grape juice; ORAC: oxygen radical absorbance capacity; OT: Olympic triathlon; Orange: possible favourable effect; P: performance; PAL: physical activity level; Ph: physiology; PHEOH: public health, environmental and occupational health; POMS: profile of mood states; PPROT: proteins of plant origin; Prdt: periodisation; Pt: proteins; PUFA, polyunsaturated fatty acids; PUFAn6: n-6 polyunsaturated atty acids; PUFAn3: n-3 polyunsaturated fatty acids; Rcv: recovery; RED-S: Relative Energy Deficiency in Sports; rED: relative energy deficit; REE: resting energy expenditure; rEI: relative energy intake; RER: respiratory exchange rate; RPE: rate of perceived exertion; rTEE: relative total energy expenditure; Red: negative effect; s: speed; S: subjective/observational; SFA: saturated fatty acids; SL: sleep low; SM: sport medicine; SNKQ: Sports Nutrition Knowledge Questionnaire; Spp: supplementation; SS: sports supplements; ST: Sprint triathlon; St: strength; STR: sport therapy and rehabilitation; T: testosterone; Ta: Taurine; TBW: total body water; Texh: time to exhaustion; TEE: total energy expenditure; TL: train low; TNF-α: cytokines; TRX: Thioredoxin; TT: Thirty-minute time trial; UE: ultra-endurance; V: volume; VAT: visceral adipose tissue; vC: vitamin C; vE: Vitamin E; VO_2max_: maximum oxygen consumption; VO_2_: oxygen consumption; VT: ventilatory threshold; W_VT_: workload expressed in watts; Yellow: possible improvement; YP: young people; Zn: zinc.

## Data Availability

The information related to this paper is not publicly available, but you can get it by contacting the author responsible for this work.

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
