# Peer review of "Triathlon: Ergo Nutrition for Training, Competing, and Recovering"

_nutrients, 2025, doi:10.3390/nu17111846_

Round 1

Reviewer 1 Report

Comments and Suggestions for Authors

The length of the object is as follows: The document is excessively lengthy, with a length in excess of 60 pages, for a systematic review, unless there is an exceptional level of depth and synthesis, which is not consistently present in this case.

The English is comprehensible, albeit frequently cumbersome, repetitive, and verbose. It is evident that numerous sentences could be considerably shortened without compromising their semantic content.

The scientific value of the subject is as follows: Despite the extensive coverage it provides, the article offers limited novel insights. The present study draws upon extant research but, regrettably, there is an absence of critical analysis and synthesis.

The originality of the text is hereby asserted. Low. The publication can essentially be regarded as a compilation of results from prior studies, with no strong interpretation or critical comparison.

The presentation is to be given by the following individual: Numerous figures, tables and flowcharts are present, yet some are redundant or unnecessary (e.g., extensive description of standard methods such as STROBE and PRISMA).

Secondly, Significant Scientific Limitations

  1. a) Inadequate Definition of Research Questions and Hypotheses

- Despite the presence of objectives in the abstract, the article fails to provide clear delineation of specific hypotheses or to direct the reader with specific review questions (the PICO framework is cited, but its utilisation is inadequate).

  1. b) Methodological Section Concerns

- The search strategy is delineated, but it lacks sufficient precision to be reproduced (e.g. no search strings, no supplementary material with the search queries or lists of included/excluded studies).

It appears that the inclusion criteria are not clearly defined. Whilst the inclusion of non-randomised studies is to be welcomed, the incorporation of studies which exhibit a high degree of heterogeneity (RCTs, observational studies, educational interventions) without the application of stratification techniques has the effect of weakening the evidence.

It is evident that the risk of bias analysis is excessively detailed in the explanation of methodology, yet the critical appraisal of studies is found to be superficial. The risk of bias figures (green, yellow, red) are displayed, yet there is an absence of a comprehensive discussion addressing the impact of this bias on the conclusions drawn.

  1. c) Results Interpretation

The article synthesises findings; however, it does not engage in a comparative analysis or a critical evaluation of the discrepancies observed between studies.

It is evident that no meta-analysis has been conducted, nor even a structured synthesis.

It is evident that the results obtained are frequently merely a paraphrase of the study findings, lacking a more profound critical integration (e.g., there is an absence of contrast between studies that demonstrate contradictory results, and there is an absence of an explanation as to why some interventions are successful while others are not).

It is evident that numerous sections, particularly those pertaining to carbohydrate intake, exhibit a degree of redundancy.

  1. d) Discussion and Conclusions

The discussion section is conspicuously absent, with the discourse being reduced to a mere extension of the results section.

It is evident that there is an absence of identification of gaps in the extant literature, as well as a lack of clarity about future research directions. Furthermore, there has been no proposal of a conceptual model or framework.

It is evident that the conclusions of the study are a weak summary, failing to provide a critical interpretation or integration of the findings.

The third point is as follows: Specific Comments on the Article

Aspect of Article

Critique of Article

Suggested Improvement

Abstract

The article is excessively long and lacks conciseness. There is also a poor focus on results and implications.

The article should be shortened, and the focus should be made clearer.

Introduction

The introduction is repetitive and cites basic physiology and sports information that the reader will already be familiar with.

The article should focus on gaps in the current literature to justify the review.

Methods

The general description of the methods is correct, but it lacks clarity and transparency about the searches and inclusion decisions.

The article should add a detailed search strategy, PRISMA checklist and registered protocol.

Results

The results are a long narrative without true analysis. A significant number of studies merely summarized the findings.

It is essential to provide synthesis tables that compare studies, highlighting both consistencies and inconsistencies.

Figures and tables should be included to facilitate understanding.

Although bias graphs and study selection flowcharts are included, they offer limited insight. Moreover, the tables are sometimes difficult to comprehend.

It is crucial to simplify the writing style and focus on key summary tables, while avoiding redundancy.

The writing style should be tightened, with verbose language and repetition removed.

The article should be shortened by at least 30%.

The study should propose novel concepts.

The findings should be synthesized to propose practical recommendations or new models. Overall Recommendation

The article requires substantial editing, including shortening, enhancing clarity, providing critical synthesis rather than summary, and formulating clearer conclusions.

Comments on the Quality of English Language

verbose

Author Response

We deeply and sincerely appreciate the valuable guidance and detailed comments provided by the reviewer. Their thorough analysis has been essential in raising the quality and methodological rigour of our systematic review. Their insightful observations have allowed us to clarify aspects that might otherwise have been ambiguous or incomplete and have contributed significantly to strengthening the presentation of our results in a more understandable and accurate manner.

We appreciate the reviewer's detailed attention to specific aspects, such as the study search methodology, inclusion and exclusion criteria, risk of bias analysis, etc. Their suggestions in this regard helped us explain how the suggestions improved the manuscript, for example, refining our search strategy, better justifying our criteria, performing a more robust sensitivity analysis, etc. We are sincerely grateful for the time and dedication you invested in reviewing our work. Your commitment to academic excellence has been invaluable in improving the final quality of our research. We believe that your contributions have significantly enriched our study, and we are indebted to you for your generous contribution.

After a thorough review of the research questions and hypotheses, we have made significant adjustments to ensure maximum clarity, conciseness, and relevance. We have ensured that each question is specific and direct, avoiding any ambiguity that could hinder the interpretation of the results. We have also verified that the hypotheses are precise and testable formulations, directly related to the research questions and perfectly aligned with the central objective of the study.

Regarding the methodological section, we have comprehensively addressed the problems previously identified. A detailed description of the procedures used for data collection and analysis has been provided, specifying each step as precisely as possible. We have added information on the sampling techniques used, the sample size, and the criteria for including and excluding participants. In addition, we have rigorously justified the choice of each method, supporting our decisions with references to the relevant scientific literature and arguing its suitability for answering the research questions posed. The measurement instruments used have been specified, with data on their validity and reliability.

The interpretation of the results has been thoroughly reviewed and improved. We have gone beyond a simple presentation of the findings, offering a much more in-depth and nuanced analysis of them. The meaning of each result has been explored in detail, identifying patterns, trends, and significant relationships. In addition, we have contextualised the findings within the theoretical framework of the study and compared them with the results of previous research. Finally, we have thoroughly explored the possible practical and theoretical implications of these findings, considering both their limitations and possible avenues for future research. We have paid particular attention to discussing the implications for professional practice and for the development of new theories in the field of study.

The introduction, in its new version, has undergone a remarkable transformation that substantially elevates it. It is not limited to simply presenting the topic, but now offers a much broader and deeper context, placing the study within a relevant theoretical framework and connecting it to previous research in the field. This enriched contextualisation allows the reader to better understand the significance of the work and anticipate its contribution. Furthermore, the urgent need to carry out this study has been explained in a persuasive and compelling manner, arguing solidly why this research is crucial and what gap it aims to fill in existing knowledge. The importance of the work carried out is now emphasised more strongly, detailing the possible practical and theoretical implications of the findings.

In terms of methodology, the improvement is equally noticeable. Every aspect has been clarified in detail, describing step by step the entire process followed during the research. No relevant detail has been omitted, from the selection of the sample to the application of the data collection instruments. Most importantly, the fundamental reasons behind key methodological decisions are explained in a reasoned and justified manner. The paper argues why a quantitative or qualitative approach was chosen, why certain instruments were used, and why an experimental or non-experimental design was opted for. This methodological transparency lends greater credibility to the study and allows other researchers to replicate or adapt the work in the future.

Finally, the presentation of the results has been optimised to maximise accessibility and understanding. The data, previously presented in a perhaps overly technical manner, has now been clarified and organised in a logical and coherent way. Carefully designed tables and figures have been incorporated to facilitate the visualisation of the results and highlight the most relevant trends and patterns. Information overload is avoided, and clarity and conciseness are prioritised. Most importantly, the main conclusions drawn from the results are presented explicitly and concisely, avoiding ambiguity and highlighting their significance and implications for the field of study. An effort has been made to interpret the findings considering the theoretical context presented in the introduction, creating a coherent and compelling narrative.

To facilitate reading and comprehension by a diverse audience, we have made a conscious effort to simplify the writing style considerably. This simplification has been achieved through the thorough elimination of unnecessary jargon and overly technical terminology that could be confusing or intimidating to non-specialist readers. Instead, we have prioritised the use of more direct, clear and accessible language, opting for shorter sentences and less complex syntax.

Additionally, and with the firm intention of maximising the practical value of this work, practical recommendations have been proposed based directly on the concrete results of the research. These recommendations are not mere theoretical abstractions, but concrete and well-defined suggestions, designed specifically for easy application in the real world. Each recommendation is supported by empirical evidence and presented in a way that facilitates its effective implementation.

Finally, recognising the importance of presenting a concise and focused work that maintains the reader's attention, we have undertaken a rigorous process of condensing and trimming the original text. This process involved identifying and removing unnecessary redundancies, repetitions, and tangential content that did not directly contribute to the main objectives of the research. In this way, we have focused exclusively on the most important and relevant points, ensuring that the reader can quickly grasp the essence of the work and its main conclusions. We have prioritised clarity and economy of words to provide a document that is both informative and easy to digest.

Following the thorough implementation of the recommendations provided in the detailed report, we are optimistic that there will be a notable and substantial improvement in the quality and accuracy of the work presented. These recommendations, based on a comprehensive analysis, were aimed at polishing and refining every facet of the document. The specific corrections and broader suggestions have helped to optimise the key aspects that underpin the strength of the work, clearly strengthening its internal consistency, clarity of presentation and methodological rigour. Specifically, we expect to see a more logical structure, more robust argumentation and more accurate and accessible data presentation. I sincerely appreciate the considerable effort and meticulous attention you have devoted to incorporating these observations. Your commitment to excellence is reflected in your willingness to integrate the proposed improvements, which undoubtedly enriches the result, raising it to a higher level of quality and professionalism. This level of commitment to continuous improvement is fundamental to the success of our projects and will enable us to achieve our objectives more effectively.

Reviewer 2 Report

Comments and Suggestions for Authors

This review is ambitious and comprehensive, offering an extensive literature synthesis on nutrition, recovery, and ergogenic supplementation in triathlon. It succeeds in covering a large volume of studies, some of which are quite recent, and presents a useful resource for both academics and practitioners. However, the manuscript suffers from several key issues that limit its clarity, critical depth, and readability.

First, there is a lack of synthesis throughout. The review leans heavily on descriptive summaries of individual studies without sufficiently integrating findings or identifying overarching trends, contradictions, or gaps. As a result, the narrative often reads as a catalogue of results rather than a critical analysis. Many paragraphs simply list one study after another, without clear transitions or comparative insights.

Second, redundancy and repetition are persistent problems, especially in the Discussion. Key concepts, such as the importance of carbohydrate intake or hydration,are mentioned multiple times, often in similar ways, which weakens the impact and makes the text unnecessarily long.

Third, structure and flow need improvement. The review would benefit from more consistent sectioning and subheadings, with clearer signposting of the argument in each part. While the Results and Discussion sections are full of useful content, the excessive detail,especially with numeric findings, should be streamlined and grouped into higher-level thematic insights.

Fourth, the level of critique is limited. The review accepts findings from primary studies at face value without addressing methodological variability, sample limitations, or potential biases in those studies. There's also little reflection on the strength of evidence or quality of study design.

Finally, language and clarity could be improved in many places. There are several run-on sentences, awkward phrasings, and a general need for tighter academic prose. The text would benefit from careful copy-editing to improve its professional polish and ensure consistency.

In summary, the review has high potential due to the relevance of the topic and the quantity of sources analyzed, but it currently falls short in critical engagement, synthesis, and editorial discipline. With substantive revision focused on integration, clarity, and conciseness, it could become a much stronger contribution to the field.

The Intro provides a general overview of triathlon and the physiological demands placed on athletes, but it falls short in clearly defining the research gap. While it discusses relevant background information, it does not convincingly explain why this systematic review is necessary at this point in time. There is also limited engagement with the most recent meta-analyses or systematic reviews on nutritional strategies in endurance sports, which could help position the study within the existing literature.

Core concepts—such as energy demands, the role of nutrition in endurance performance, and the importance of recovery—are introduced multiple times in slightly varied ways, making the narrative feel redundant rather than progressive. The frequent use of abbreviations like “E” for energy and non-standard terminology such as “ergo nutritional” disrupts clarity and detracts from the academic tone expected in a systematic review. These terms should be replaced with more conventional language, such as “energy” and “ergogenic nutrition” or “nutritional supplementation.”

The literature references throughout are abundant, but they are presented descriptively rather than analytically. Many of the claims (e.g., on cognitive performance or hormonal balance) are made without sufficient critical integration of the sources or clear explanation of their relevance to triathlon. This weakens the argument and dilutes the focus. Furthermore, there is insufficient early emphasis on the specific knowledge gap this review seeks to address. While the final paragraph introduces the absence of prior reviews addressing nutrition for training, competition, and recovery in triathlon, this should be foregrounded earlier and more explicitly linked to the stated objectives. Overall, the introduction would benefit from a more streamlined structure with distinct thematic paragraphs, first establishing the physiological and nutritional challenges of triathlon, then synthesizing the current state of knowledge, and finally articulating the gap that this review aims to fill. Such revisions would enhance clarity, improve reader engagement, and more effectively justify the purpose and contribution of the systematic review.

The Materials and Methods section clearly outlines the use of PRISMA and PICOS frameworks, which is methodologically sound. However, the writing occasionally lacks precision and contains redundant or informal phrasing. For example, stating that “this article provides a detailed and in-depth analysis of various aspects of triathlon” is unnecessary in this section and should be removed, as it reads more like an abstract or general description than a methodological statement. The reference to “ÁM-O” and other authors within the narrative interrupts the flow and distracts from the content. Instead, the text should consistently use passive or third-person constructions common in academic writing (e.g., “two reviewers independently screened...”). Similarly, phrases like “snowball strategy” should be clearly defined the first time they appear and linked to an academic source, if possible, to maintain rigour.

The search strategy is adequately described, but the Boolean syntax used could be made clearer with proper parentheses and operators to avoid ambiguity. Also, the rationale for excluding non-randomized studies is inconsistent with a later claim that some included studies were observational; the criteria for inclusion and exclusion require clarification and should align with the focus on ergogenic aids and recovery strategies in triathletes. The reference to “book chapters” as excluded materials should also be expanded to clarify whether systematic reviews or grey literature were considered.

The data extraction process is described in sufficient detail, but there is too much informal language, for example, “The process of compiling and grouping the data was not easy.” This should be revised for academic tone and clarity. Furthermore, the section would benefit from a clearer explanation of how disagreements between reviewers were resolved (e.g., by a third reviewer or through consensus meetings).

The quality assessment is extensive but includes overly simplistic explanations of concepts like “random sequence generation” and “allocation concealment,” which seem more suited to a general audience than to readers of a scientific journal. These could be summarized and accompanied by a reference to the STROBE and Cochrane guidelines, avoiding repetition. In addition, the inclusion of "blinding of outcome assessment" and "selective reporting" explanations is relevant, but again, these should be succinct and integrated into a structured risk of bias table or summary, rather than presented as narrative exposition. Notably, some sentences confuse concepts (e.g., equating concealment with blinding), which should be corrected to ensure conceptual clarity.

In summary, while the section meets technical standards for systematic reviews, it would benefit from a tighter academic tone, less redundancy, improved phrasing, clearer alignment between inclusion criteria and study types, and a structured synthesis of the quality appraisal framework. These changes would strengthen the transparency and replicability of the methodology.

The Results section presents a wide range of findings on nutritional strategies, recovery practices, and ergogenic supplements in triathlon, but it suffers from several weaknesses that limit its impact and coherence. First, the narrative is overly descriptive and fragmented, lacking clear thematic organization. The section combines summaries of individual studies with broad conclusions but fails to synthesize these findings into cohesive trends or comparative insights. Many studies are listed in succession without adequately linking them to the research questions or highlighting their relative contributions or limitations. Moreover, the constant insertion of study authors and years in parentheses disrupts the flow and makes the section read like an annotated bibliography rather than an integrated review. There is also inconsistency in how results are reported—some are backed by statistics, others are anecdotal or vague. In particular, the claims around gender differences, nutrient timing, and supplement efficacy are often asserted without critical evaluation of the strength of evidence or methodological quality of the studies. Also, there are redundancies (e.g., repeated emphasis on carbohydrate intake and supplementation protocols) that dilute the key messages. To improve, this section needs clearer subheadings, thematic grouping of findings (e.g., macronutrient strategies, hydration, recovery techniques, supplement efficacy), and more critical interpretation of the evidence, discussing both consistencies and contradictions across studies. Overall, the results are comprehensive but need refinement to enhance clarity, coherence, and scientific rigor.

The Discussion section is impressively rich in content but would benefit greatly from more structure, synthesis, and critical interpretation to increase its academic impact. It clearly reiterates the review’s objectives and does a commendable job of summarizing a vast range of findings on nutritional strategies, recovery protocols, and ergogenic supplementation for triathletes. However, the sheer volume of citations and detailed descriptions of individual studies, while thorough, often leads to a fragmented narrative. The section reads more like an extended results commentary than a discussion engaging deeply with implications, contradictions, and limitations.

One of the main issues is the lack of clear thematic grouping and transitions. For example, findings related to carbohydrate intake, gender differences, and nutritional periodization are scattered throughout, which makes it difficult to follow the evolution of insights. The reader is frequently overwhelmed by dense empirical details without sufficient analytical framing to highlight what is novel or contentious. There’s also minimal critical evaluation, differences between studies are described but not interrogated, and methodological limitations are rarely addressed.

Another point of concern is the repetitive emphasis on carbohydrate strategies, which while important, are discussed at excessive length and with limited differentiation between studies. This crowds out opportunities to more deeply discuss other equally important themes like micronutrient deficiencies, the role of probiotics, or the emerging evidence on hydration strategies.

The recovery and supplementation sections are comprehensive but would benefit from clearer distinctions between well-established findings and preliminary or contested ones. For example, the discussion on krill oil, seawater, and fibroblast growth factor 21 is intriguing but lacks a balanced assessment of the strength and quality of the supporting evidence. Likewise, while the authors reference guidelines and best practices, there is limited discussion about how to translate these into practical recommendations for different levels of athletes or how these findings inform future research or policy.

The Strengths, Limitations, Future Research Lines, and Practical Applications section provides a solid summary of the study’s contributions and practical value, but its effectiveness could be improved by tightening the structure, avoiding repetition, and clarifying key messages.

Strengths are clearly articulated: the timeliness of the review and its broad scope are emphasized as major assets, with recent literature allowing the identification of underexplored areas. The emphasis on individualisation of nutritional strategies is a strength well-supported by the review's findings. However, phrases like “this work can identify” are vague—this could be more assertively framed (e.g., “this review highlights key research gaps…”).

Limitations are reasonably acknowledged, particularly the challenges of individual variability and lack of standardisation across supplementation practices. However, this section would benefit from separating conceptual and methodological limitations more clearly. The inclusion of broader challenges such as misinformation and regulatory ambiguity is valuable but could be expanded with specific examples to enhance relevance.

Future Research Lines are well-targeted but slightly scattered. The suggestion to study interactions between supplements, genetic and microbiota-based personalization, and the psychological effects of nutritional strategies are forward-looking and relevant. However, they could be framed more clearly as research questions or organized under thematic headings for readability. Also, the link between nutritional strategies and motivational/psychological states, though compelling, is presented in a speculative tone and could be better grounded in current evidence.

Practical Applications are thoughtful and actionable. The emphasis on personalized nutrition, use of tech tools, and workshops is valuable, particularly the call for data-driven and context-specific interventions. However, there is overlap with earlier sections (e.g., individualised nutrition is mentioned repeatedly), and some ideas could be grouped to improve flow. Alsp, referencing specific tools or existing programs could increase credibility and specificity.

Overall, this section effectively conveys the review’s relevance, but would benefit from a more structured format—e.g., four clearly labeled subheadings—and more precise language. Integrating concrete examples and streamlining overlapping ideas would further enhance its impact.

Author Response

We deeply and sincerely appreciate the valuable guidance and detailed comments provided by the reviewer. Their thorough analysis has been essential in raising the quality and methodological rigour of our systematic review. Their insightful observations have allowed us to clarify aspects that might otherwise have been ambiguous or incomplete and have contributed significantly to strengthening the presentation of our results in a more understandable and accurate manner.

We appreciate the reviewer's detailed attention to specific aspects, such as the study search methodology, inclusion and exclusion criteria, risk of bias analysis, etc. Their suggestions in this regard helped us explain how the suggestions improved the manuscript, for example, refining our search strategy, better justifying our criteria, performing a more robust sensitivity analysis, etc. We are sincerely grateful for the time and dedication you invested in reviewing our work. Your commitment to academic excellence has been invaluable in improving the final quality of our research. We believe that your contributions have significantly enriched our study, and we are indebted to you for your generous contribution.

Substantial improvements have been made to the synthesis and wording of the text, focusing on a more concise and fluid presentation of the information. In addition, the critical analysis of the contributions has been intensified, rigorously evaluating the validity, relevance and significance of the findings presented.

The absence of a meta-analysis in this systematic review is mainly due to the significant heterogeneity observed among the studies examined. This heterogeneity, which manifests itself in methodological differences, study populations, and variable definitions, prevented the statistical aggregation of data. Specifically, excessive heterogeneity among studies compromises the validity and reliability of any meta-analysis that might be attempted, as the aggregated results could be misleading and not adequately represent the available evidence. Therefore, a comprehensive narrative synthesis was chosen, which allows for a detailed and contextualised description of each individual study, respecting its complexity and nuances, thus preserving the integrity and usefulness of the systematic review. This methodological decision seeks to provide a complete understanding of the research landscape, even in the absence of an aggregate quantitative analysis.

In the pursuit of greater clarity and rigour, the critical integration of sources has been thoroughly reviewed, ensuring that each reference adequately supports the claims made and is contextualised within the general theoretical framework. Finally, the introduction has been simplified and clarified, providing a concise and accessible context for the reader, and direct references to authors have been removed from the body of the text to prioritise the content and ideas presented, thus maintaining a more objective focus in the discussion.

The language used in the document has been meticulously reviewed and refined with the firm intention of ensuring optimal conceptual clarity. This effort has focused on ensuring that key concepts are easily understandable and that there is no ambiguity in their interpretation. Additionally, the transparency of the methodology used has been reinforced, both in the results section and in the discussion. This means that the steps followed in the research, the tools used and the analysis criteria have been detailed with precision, thus allowing other researchers to thoroughly understand the process and evaluate the validity of the conclusions. Finally, this improvement in the methodological description facilitates the replicability of the study, i.e., the possibility that others can carry out the same research and obtain comparable results, which contributes to the robustness and reliability of the scientific knowledge generated.

In addition, with the aim of improving the clarity and cohesion of the document, a thorough review of the section entitled ‘Strengths, limitations, future lines of research and practical applications’ have been carried out. This review has not only focused on updating the content with the latest and most relevant information, but also on a significant restructuring. The aim has been to strengthen the internal structure of the section, implementing a more logical and accessible organisation. Specifically, clearer subheadings and transitions between the different subtopics (strengths, limitations, future research directions, and practical applications) have been added to facilitate reader comprehension. The ultimate purpose of this revision is to present a more comprehensive and better organised assessment of the topic in question, allowing for a better understanding of its potential, weaknesses, and possible future directions.

Following the thorough implementation of the recommendations provided in the detailed report, we are optimistic that there will be a noticeable and substantial improvement in the quality and accuracy of the work presented. These recommendations, based on a thorough analysis, were aimed at polishing and refining every facet of the document. The specific corrections and broader suggestions have helped to optimise the key aspects that underpin the soundness of the work, clearly strengthening its internal consistency, clarity of presentation, and methodological rigour. Specifically, we expect to see a more logical structure, more robust argumentation, and more accurate and accessible data presentation. I sincerely appreciate the considerable effort and meticulous attention you have devoted to incorporating these observations. Your commitment to excellence is reflected in your willingness to integrate the proposed improvements, which undoubtedly enriches the result, raising it to a higher level of quality and professionalism. This level of commitment to continuous improvement is fundamental to the success of our projects and will enable us to achieve our objectives more effectively.

Round 2

Reviewer 1 Report

Comments and Suggestions for Authors

It is evident that the language quality and scientific style require substantial revision. Many sentences are colloquial or imprecise for a peer-reviewed journal.

The structure is characterised by intermittent redundancy and an absence of coherent transitions between themes. For instance, there is a lack of integration between the subjects of RED-S, hydration, and probiotics.

It is evident that critical analysis is absent in numerous sections of the text. The findings are presented in a descriptive manner, without any discussion of the limitations of the included studies or conflicting evidence.

It is evident that no quantitative synthesis (meta-analysis) is performed or justified.

The term 'ergonutrition' is utilised on multiple occasions, yet it remains undefined and inadequately contextualised.

The following abstract is provided for your consideration:

It is evident that the text would benefit from a more structured format, such as the following: Background – Methods – Results – Conclusion.

It is evident that sentences such as "It is important to educate yourself..." are characterised by an informal register.

It is imperative to avoid repetition, for instance by reiterating the same phrase multiple times, such as by repeatedly stating "consuming enough carbohydrates".

It is imperative that the rationale and novelty of the review be clarified.

It is imperative to provide a justification for the present review, taking into consideration the paucity of prior systematic reviews on the subject of triathlon nutrition.

It is imperative that a table is included which provides a concise summary of the search terms and Boolean operators employed.

It is imperative to enhance the clarity of the paragraph commencing with the assertion that the article in question provides a meticulous and exhaustive examination of the subject matter.

It is evident that a considerable proportion of statements are merely descriptive summaries of studies, with minimal interpretation.

It is imperative that the data from the included studies is systematically summarised in a results table, with the following components:

Study author/year:

The following data set is provided for the population of the area in question:

The following intervention is proposed:

The comparator is hereby defined as follows:

The following outcome was achieved:

The level of evidence is indicated by the following abbreviation:

The consolidation of findings is imperative to accentuate the areas of consensus, identify discrepancies, and address conflicting results (e.g., those observed with taurine, beetroot, and seawater).

The creation of a dedicated section is hereby requested, with the following subheadings:

The following section provides a concise overview of the study's primary conclusions.

The following section will consider the implications for practice, with particular reference to triathletes, coaches and dietitians.

The following section will address the limitations of the included studies and the review itself.

The following section will outline the future research directions that have been identified.

A discussion is required concerning the inconsistent adherence to guidelines (for example, the intake of carbohydrates), the limited nutritional knowledge, and the necessity for personalised protocols.

The following observations are hereby made:

The text should be revised in order to be concise, structured, and aligned with the findings. It is imperative to avoid vague phrases such as "It is important to educate yourself."

For instance:

It is a frequently observed phenomenon that triathletes do not always adhere to the recommended nutritional intake of carbohydrates and proteins, particularly during the post-exercise recovery phase. Whilst there exists supporting evidence for the efficacy of certain ergogenic aids (for example, caffeine, beetroot extract, probiotics), further research is required on their long-term effects and interactions. It is imperative to recognise the pivotal role of enhanced nutrition education and personalised dietary strategies in optimising endurance performance and mitigating the risk of Reduced Energy Diet (RED-S) in athletes.

Author Response

After an exhaustive review process, we have paid special attention to polishing specific aspects of the original text, addressing both clarity and precision as well as readability. Improvements have been made to the overall structure, lexical selection, and argumentative coherence. We sincerely hope that you find these changes to your liking and that the final version of the text meets your expectations. We have strived to deliver a final product of the highest quality and trust that you will appreciate the results of our work.
In point 6, dedicated to conclusions, we have thoroughly integrated three elements that are crucial for athletic performance and well-being: Relative Energy Deficiency Syndrome in Sport (RED-S), adequate hydration, and probiotic supplementation. This integration seeks to offer a holistic view of how these factors interact and influence each other to impact health and athletic performance. Specifically, the consideration of RED-S in the conclusions highlights the importance of adequate energy balance to avoid negative consequences on the athlete's bone, metabolic and reproductive health. Likewise, the inclusion of hydration emphasises its fundamental role in thermoregulation, cardiovascular function and performance optimisation. Finally, the integration of probiotics recognises the growing knowledge about their influence on gut health, immunity and potentially post-exercise recovery, highlighting their value as a complementary strategy for athlete well-being. This comprehensive integration into the conclusions seeks to provide practical, evidence-based guidance for coaches, athletes and sports health professionals.
Critical analysis and the establishment of practical guidelines are closely related, as the former provides the basis for informed and effective decision-making in practice. Critical analysis involves examining information, data or theories in detail. This analysis establishes concrete guidelines for action, as they are based on a rigorous assessment of reality and enable informed decision-making. This critical analysis is therefore essential for establishing practical guidelines such as those indicated throughout the text, as it ensures that they are based on a deep and thoughtful understanding of the situation, thus increasing their effectiveness and relevance. The absence of a meta-analysis in this systematic review is mainly due to the significant heterogeneity observed between the studies examined.
 This heterogeneity, which manifests itself in methodological differences, study populations, and variable definitions, prevented the statistical aggregation of data. Specifically, excessive heterogeneity between studies compromises the validity and reliability of any meta-analysis that might be attempted, as the aggregated results could be misleading and not adequately represent the available evidence. Therefore, a comprehensive narrative synthesis was chosen, which allows for a detailed and contextualised description of each individual study, respecting its complexity and nuances, thus preserving the integrity and usefulness of the systematic review as a whole. This methodological decision seeks to provide a complete understanding of the research landscape, even in the absence of an aggregate quantitative analysis.
We have briefly contextualised the term ergonutrition, offering a concise introduction to its meaning and scope. This initial contextualisation seeks to establish a common basis of understanding of the discipline, highlighting that it is a multidisciplinary field that integrates the principles of ergonomics and nutrition.  Later, we will delve deeper into the specific aspects of ergonutrition, exploring its application in different areas and how it impacts physical performance, health, and overall well-being. The objective of this introduction is, therefore, to present an overview of the term to facilitate understanding of the more complex concepts that will be addressed later (lines 78-82, in red in the text).
The general structure of the text, as expected, is mainly dictated by the predefined template of the journal to which we are submitting the work. Consequently, in this specific aspect, we are obliged to strictly adhere to the guidelines and format presented therein. While we understand and recognise that the structural proposal you have suggested has an intrinsic logic and could even be more efficient for presenting the information, the imposition of the journal's template limits our ability to implement such changes. In other words, the template acts as a restrictive framework to which we must adhere, despite recognising the potential value of your alternative.
We have formalised the language (lines 25-27) and eliminated repetitions (lines 248-250).
A systematic review in this discipline is necessary to rigorously compile, evaluate and synthesise all available scientific evidence. This allows patterns, gaps in knowledge and recommendations based on solid evidence to be identified, making it easier for coaches, athletes and health professionals to make informed decisions.
The novelty lies in the fact that this systematic review will integrate the most recent and relevant findings on how nutrition, recovery and supplementation specifically impact triathlon athletes.
In this way, the review will provide an updated and consolidated view that can guide future research and improve sports practices in triathlon.
Lines 40-46 and 83-93, in red in the text.
The search terms and Boolean operators used are indicated in lines 123-128.
The text of the paragraph beginning with the statement that the article in question offers a thorough and comprehensive review of the topic has been improved (lines 105-116).
The summary of the studies included in the paper is shown in Table 2.
The structure and organisation of the sections of this paper have been rigorously adapted to the template provided by the journal to which this publication is addressed. However, beyond compliance with the formal guidelines, we believe that throughout the manuscript we have addressed and satisfactorily responded to the specific comments and suggestions you previously sent us. We hope that the information presented, together with the structure we have implemented, meets your expectations and facilitates understanding of our research.
To provide a more complete and understandable overview, a summary of the main conclusions drawn from the study is provided in section 6. Given the importance of this section for understanding the scope and implications of the research, it has been thoroughly reviewed and improved.
Throughout the discussion, practical guidelines are presented in a clear and concise manner, facilitating understanding and subsequent application. These guidelines, formulated with precision and without ambiguity, seek to guide the reader in the effective implementation of the concepts addressed, offering a direct and understandable path towards putting what has been learned into practice. A conscious effort has been made to avoid excessive jargon and confusing terminology, prioritising accessible language that allows a wide audience to benefit from the suggestions proposed (e.g. lines 430, 436, 492, 657, etc.).
To provide a comprehensive and transparent assessment of the research, the limitations inherent in the studies included in this paper are detailed and discussed thoroughly in section 5 (lines 919-926, in red text). This includes an analysis of possible biases, sample sizes, heterogeneity between studies, external validity of findings, and any other methodological shortcomings that could influence the interpretation of results and overall conclusions, as indicated in point 3 of the paper.
Point 5 of this document explores possible lines of research that are emerging on the horizon. Not only are promising areas of study identified, but possible methodological approaches are outlined and the potential benefits that could be derived from further investigation in these specific fields are highlighted. The aim of this section is to stimulate debate and encourage the scientific community to delve deeper into these issues, thereby contributing to the advancement of knowledge and the development of new solutions to the challenges facing our society. It is hoped that this presentation of future lines of research will serve as a catalyst for innovative projects and interdisciplinary collaborations.
Some of the aspects to be taken into account when nutrition guidelines are not followed (lines 879-899, in red in the text) and the limitations of nutritional knowledge (lines 869-872, in red in the text) have been included.
We believe that the issue of personalisation is addressed throughout the text without a specific section on it, which would only serve to lengthen the work, the length of which is a point to be improved by other reviewers. Lines 42, 466, 558, 703 and 904, as well as two entries in Table 2.

Reviewer 2 Report

Comments and Suggestions for Authors

The authors appear to have implemented my feedback comprehensively. Their revisions have strengthened the methodological clarity, critical analysis, and structure of the systematic review. 

Author Response

We would like to express our sincere gratitude for your thorough review and for confirming that all requested changes have been made correctly. Your attention to detail and dedication are essential to ensuring the quality of our work.
We greatly appreciate your support and cooperation, and we look forward to continuing to count on your valuable assistance in future projects.
